# ASCIIEval: Benchmarking Models' Visual Perception in Text Strings via ASCII Art

**Qi Jia**[1]    **Xiang Yue**[3]    **Shanshan Huang**[4]    **Ziheng Qin**[2]    **Yizhu Liu**[5]
**Bill Yuchen Lin**[6]    **Yang You**[2†]    **Guangtao Zhai**[1,7†]

[1]Shanghai Artificial Intelligence Laboratory    [2]National University of Singapore
[3]Carnegie Mellon University    [4]Guangzhou University    [5]Meituan
[6]University of Washington    [7]Shanghai Jiao Tong University

✉ jiaqi@pjlab.org.cn, youy@nus.edu.sg, zhaiguangtao@sjtu.edu.cn

## Abstract

Perceiving visual semantics embedded within consecutive characters is a crucial yet under-explored capability for both Large Language Models (LLMs) and Multi-modal Large Language Models (MLLMs). In this work, we select ASCII art as a representative artifact. It depicts concepts through careful arrangement of characters, which can be formulated in both text and image modalities. We frame the problem as a recognition task, and construct a novel benchmark, ASCIIEval. It covers over 3K samples with an elaborate categorization tree, along with a training set for further enhancement. Encompassing a comprehensive analysis of tens of models through different input modalities, our benchmark demonstrate its multi-faceted diagnostic power. Given textual input, language models shows their visual perception ability on ASCII art concepts. Proprietary models achieve over 70% accuracy on certain categories, with GPT-5 topping the rank. For image inputs, we reveal that open-source MLLMs suffer from a trade-off between fine-grained text recognition and collective visual perception. They exhibit limited generalization ability to this special kind of arts, leading to the dramatic gap of over 20.01% accuracy compared with their proprietary counterparts. Another critical finding is that model performance is sensitive to the length of the ASCII art, with this sensitivity varying across input modalities. Unfortunately, none of the models could successfully benefit from the simultaneous provision of both modalities, highlighting the need for more flexible modality-fusion approaches. Besides, we also introduce approaches for further enhancement and discuss future directions. Resources are available at `https://github.com/JiaQiSJTU/VisionInText`.

## 1 Introduction

While conventional wisdom suggests that texts primarily function as carriers of linguistic information and images as conveyors of visual information, real-world scenarios often involve the integration of multiple information formats. For example, images may carry textual information, thus Optical Character Recognition (OCR) (Mori et al., 1992) has been extensively studied. It focuses on capturing and understanding linguistic information embedded in images through visual processors, which is a crucial ability required in modern models for visual reasoning tasks (Liu et al., 2024b). In contrast, the comprehension of visual information embedded within text strings has not received commensurate attention.

Upon pre-training on a vast amount of text corpus, language models are generally hypothesized to be capable of capturing 2D structures in human writtings through escape characters, such as "\n". However, they were predominately assessed via textual-semantic-based benchmarks, without focused analysis on their visual perception ability. Understanding how well models can capture visual semantics in text strings is valuable for both academic research and practical applications. A

---

†Corresponding authors.

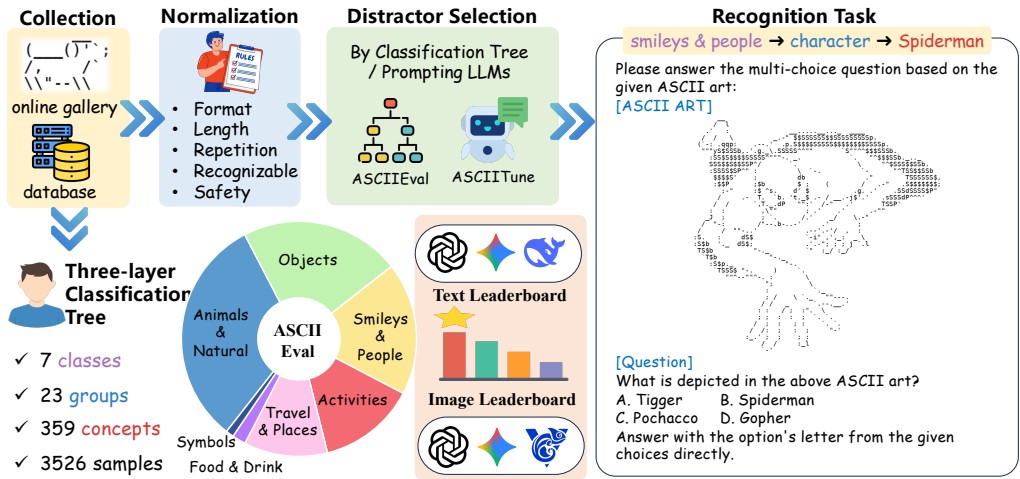

Figure 1: Overview of the ASCIIEVAL Benchmark.

natural and representative choice is ASCII art(Xu et al., 2016) as shown in Fig. 1. Visual information in these artifacts is situated in the middle of text strings and images, and can be readily expressed in both formats containing identical content. In other words, it is modality-agnostic and therefore emerges as an ideal tool for benchmarking LLMs' visual perception ability.

As for MLLMs (Achiam et al., 2023; Reid et al., 2024; Anthropic, 2024) that arm LLMs with visual processors, the character-based nature of ASCII arts presents a unique challenge. Its visual style differs starkly from images in standard benchmarks, thereby providing a rigorous test of the MLLMs' visual generalization ability. Beyond generalization, the inherent modality-agnostic quality of ASCII art serves as an excellent proxy for evaluating cross-modality alignment. A well-aligned MLLM is expected to not only perform robustly among different modalities, but also take the best of both worlds when two modalities are presented simultaneously.

Moreover, this research can also benefit a wide range of applications and have significant safety implication for LLMs and MLLMs. Such visual information is ubiquitous in a wide range of practical scenarios, such as processing tabular data (Deng et al., 2024), spatial reasoning (Wu et al., 2024) and playing board games (Topsakal & Harper, 2024). On the safety front, using visual information reflected in characters to break through the defense line is emerging as a vulnerability for adversarial attacks (Jiang et al., 2024b). For example, the attacker may use the ASCII art of a "bomb" instead of the word itself to circumvent safety protocols. A thorough analysis for understanding models' visual perception ability should be helpful for making proactive defense.

In this work, we investigate models' visual perception ability in text strings through ASCII arts with comprehensive evaluation and fine-tuning. Different from previous work that has focused on box diagrams (Hayatpur et al., 2024; Bayani, 2024), rich-formatting texts (Jiang et al., 2024b), or tone-based ASCII art (Wang et al., 2024a) that can be easily generated by rules or converted from images, we focus on ASCII art drawn by human artists, which is notably more abstract, replete with visual information, and popular among people. We formulate the task as a multiple-choice question-answering problem illustrated in Fig. 1, where the answers are objective for straightforward verification. Then, we task models to recognize the concept depicted in the ASCII art. Due to the lack of a dataset covering diverse categories that can thoroughly benchmark the ability of models, we collected data from different sources and cleaned manually under an elaborate categorization tree. In this way, we construct a test set dubbed ASCIIEVAL covering 359 concepts, together with a training set with approximately 10k data points.

Our benchmark assesses over 50 proprietary models and open-source models given different modalities of ASCII Art. This set of models, featuring models released from 2023 to the present, charts the generational progress of AI systems. Our major findings are summarized as follows:

○ **Language models demonstrate the ability to comprehend visual information solely from textual input.** Although performance on ASCIIEval strongly correlated with certain established

benchmarks, it introduces greater challenges and reveals a widening performance gap between proprietary and open-source models. To bridge the gap, we propose rationale-assisted fine-tuning with data distilled from superior models (Sec. 5).

○ For image inputs, our results indicate substantial room for improvement on this straightforward recognition task. We observe a notable regression where newer-generation open-source MLLMs underperform their ancestors. Further analysis identified **a trade-off between OCR and ASCII art recognition: an overemphasis on improving OCR will inadvertently impair models' ability to perceive collective visual signals**. We propose two post-hoc methods for mitigation: low-resolution prompting and supervised fine-tuning (Sec. 6).

○ Models exhibit different performance trends on ASCII art of increasing length, contingent upon the input modality. When text and image information are provided simultaneously, performance degrades. This reveals an **incapacity of current models to dynamically synthesize congruent cross-modal signals**, resulting in inter-modal interference rather than synergistic enhancement (Sec. 7).

## 2 BACKGROUNDS & RELATED WORK

### 2.1 LLM & MLLM BENCHMARKS

Current LLM evaluations primarily assess capabilities in knowledge, reasoning, and instruction following through benchmarks like MMLU (Hendrycks et al., 2021), Frontiermath Glazer et al. (2024), and Multi-IF He et al. (2024b), with visual perception remaining understudied except for recent program-based approaches (Qiu et al., 2025a). Similarly, MLLM benchmarks (MMMU (Yue et al., 2024), MMStar (Chen et al., 2024)) primarily evaluate multimodal understanding using conventional images rather than text-based visual representations. These benchmarks also lack guarantees of modality equivalence in mixed inputs, which is a key characteristic of ASCII art where text and visual semantics align.

Existing ASCII-related tasks remain limited: BigBench (Ghazal et al., 2013) includes basic character recognition tasks, while Gu et al. (2024) features only 40 varied ASCII generation samples. Current approaches often rely on automated conversions (e.g., Figlet [1]), risking model overfitting to transformation patterns rather than genuine visual understanding. Differing from previous work, we focus on ASCII art depicting real-world profiles with abstract visual features. We propose ASCII recognition as foundational to generation tasks and propose ASCIIEVAL, a dual-purpose benchmark for LLMs and MLLMs that uniquely combines semantic alignment across modalities with challenging visual abstraction.

### 2.2 RESEARCH ON ASCII ARTS

The origins of ASCII art date to the 1860s, evolving into a key graphic design technique as early computers utilized text characters for graphical simulation. While broadly encompassing styles like emoticons and animated art (Carlsson & Miller, 2012), it strictly consists of 95 printable fixed-width ASCII characters (Xu et al., 2016), ensuring cross-system consistency through textual representation. Early research focused on ASCII art extraction from texts using byte patterns and compression analysis (Hiroki & Minoru, 2005; Suzuki, 2011). Later computer vision studies established two synthesis approaches: tone-based (intensity distribution) and structure-based (content outlines), with the latter proving more challenging for automation (Xu et al., 2010; Chung & Kwon, 2022).

ASCII art classification research typically converts text graphics into images, leveraging image features to enhance deep neural network accuracy (Fujisawa et al., 2020; Matsumoto et al., 2018; Fujisawa et al., 2018). Fujisawa et al. (2020) automates ASCII art data generation to improve image classification. However, most studies rely on datasets with only five categories, limiting comprehensive analysis of LLMs' and MLLMs' visual representation capabilities. Other works explore ASCII art for specific purposes. Jiang et al. (2024b) demonstrate its effectiveness in jailbreak attacks bypassing advanced defenses by representing rich-format texts as ASCII art. Conversely, Wang et al. (2024a) show that tone-based ASCII art with rich visual details is unintelligible to current LLMs, making it useful for bot detection. Additionally, Wu et al. (2024) use ASCII art to improve LLMs'

---

[1] http://www.figlet.org/

spatial reasoning, while box diagrams—a specialized form of ASCII art—are benchmarked in tasks like recognition and generation (Hayatpur et al., 2024; Bayani, 2024).

Our work positions ASCII art as a unique modality bridge, enabling systematic evaluation of modality-agnostic visual perception ability for both LLMs and MLLMs.

## 3 ASCII ART RECOGNITION

We first define the ASCII art recognition task formally. Then, we introduced how we constructed the test and training data, dubbed ASCIIEVAL and ASCIITUNE, followed by statistical analysis.

### 3.1 PROBLEM FORMULATION

We formulate ASCII art recognition as a multiple-choice question-answering (QA) task. Let $x_{\text{text}}$ denote the raw textual representation of an ASCII art and $x_{\text{img}}$ its corresponding rendered image. The model's objective is to recognize the correct concept depicted in the ASCII art from a set of candidates, $\mathcal{C} = \{c_1, c_2, ..., c_k\}$. For a Large Language Model (LLM), which processes only textual input, the prediction $\hat{y}$ is generated as follows:

$$\hat{y}_{\text{text}} = \text{LLM}(x_{\text{text}}, \mathcal{C}) \tag{1}$$

A Multimodal Large Language Model (MLLM) can be prompted under two additional settings that leverage the visual modality:

$$\hat{y}_{\text{img}} = \text{MLLM}(x_{\text{img}}, \mathcal{C}) \tag{2}$$

$$\hat{y}_{\text{multi}} = \text{MLLM}(x_{\text{img}}, x_{\text{text}}, \mathcal{C}) \tag{3}$$

We refer to these three inference settings as Text-only, Image-only, and Text-Image, respectively. The prompt templates specified for each setting are detailed in Appendix D.

### 3.2 DATASET CONSTRUCTION

We carried out the data construction process in four stages to collect a high-quality test dataset.

**Data Collection**    We collect ASCII art created by artists from online galleries and existing datasets.

**Classification Criteria**    Next, we manually designed a *3-layer classification tree* after unifying the categories based on the categorical information from the original sources and removing potentially harmful categories. The most fine-grained category is named the **concept**, representing the semantic meaning reflected in the art. Similar concepts are merged into second-layer **groups**. Finally, they are grouped into seven major **classes** inspired by the iOS emoji categories. Each concept can be depicted in various ways by artists.

**Normalization & Filtering**    Subsequently, we conducted additional filtering operations using a combination of rules and human annotations as follows:

  ∘ Each ASCII art string was normalized by removing redundant empty spaces at the beginning of each line and at the end of the string, without compromising its visual semantics.

  ∘ ASCII art consisting of more than 100 lines, not belonging to reserved categories, and repetitive to other ASCII arts under the same concept were discarded. Repetition was identified by calculating the edit distance between two ASCII strings. If the distance divided by the length of the existing string was smaller than 0.3, the new ASCII art will be considered redundant.

  ∘ Human annotators were tasked to filter out unrecognizable or ambiguous art, remove words in ASCII art to focus the dataset on visual perception and avoid information leakage through words, and adjust the category according to the 3-layer category tree (See more analysis in Appendix F).

**Multiple-Choice Data Construction**    Finally, we collected negative choices for each ASCII art by randomly sampling from other concepts within the same group. It should be noted that the ground truth labels were initially collected from the sources and subsequently verified by human annotators during the data filtering process. Each ASCII art string was then converted into an image.

Table 1: Statistics of ASCIIEVAL and ASCIITUNE. The average token count is around 300 varied for different tokenizers (See Appendix E), respecting the context length limitation of all models.

| Dataset | #Samples | #Concepts | #Characters | | | #Lines | | |
|---|---|---|---|---|---|---|---|---|
| ASCIIEVAL | 3,526 | 359 | 4 | 15,282 | 635.53 | 1 | 100 | 16.97 |
| ASCIITUNE | 11,836 | 2,307 | 1 | 13,569 | 622.38 | 1 | 97 | 15.22 |

The training dataset ASCIITUNE is constructed in the same format requiring less human efforts. The negative choices are generated by prompting Llama-3-70B-Instruct and the unsafe samples recognized by Perspective API are filtered out. More details are shown in Appendix E.

## 3.3 DATA ANALYSIS

As shown in Table 1, ASCIIEVAL comprises 3,526 samples distributed across 359 concepts, 23 groups, and 7 classes. The data distribution is illustrated in Fig. 1 (More in Appendix F). Each concept is represented by 9.82 ASCII art pieces on average, with a maximum of 170 and a minimum of 1, indicating an imbalance. ASCIITUNE consists of 11,836 samples across 2,307 concepts, which is more diverse but of lower quality. The number of lines in ASCIIEVAL ranges from 1 to 100, reflecting its diversity and complexity. ASCIITUNE holds similar statistics.

**Human Upper Bound**   We randomly extracted 100 samples from ASCIIEVAL three times and asked three different annotators to perform the multiple-choice task. They achieved 100%, 98% and 97% accuracy, respectively, demonstrating that the simplicity of this visual perception task.

## 4 EXPERIMENT SETUP

**Evaluated Models**   We benchmark a wide range of LLMs and MLLMs released from 2023 to 2025 from different model families. For open-source instructed models, we experiment with LLMs including **Llama** (Touvron et al., 2023), **Qwen** (Bai et al., 2023a; Team, 2024b; Yang et al., 2025), **Mistral** (Jiang et al., 2024a), **Gemma** (Team, 2024a; Team et al., 2025) and **DeepSeek** Liu et al. (2024a), and with MLLMs containing **Llava** (Liu et al., 2023), **CogVLM** (Wang et al., 2024b), **Qwen-VL** (Bai et al., 2023b; 2025), and **InternVL** Zhu et al. (2025a). Besides, we selected several leading proprietary models including GPT-4o (OpenAI, 2023), GPT-5, Gemini-1.5-pro (Reid et al., 2024), Gemini-2.5-pro Comanici et al. (2025), and Claude-opus-4. More in Appendix G.

**Evaluation Metrics**   We evaluate model performance on ASCIIEVAL using accuracy, determined by an exact match between the model's output and the correct option. As detailed in Sec 3.3, the dataset exhibits a significant class imbalance across concepts. Therefore, we adopt *macro-average* over each concept for quantifying model performance, and *micro-accuracy* over each sample for analyzing specific ASCII art characteristics.

## 5 BENCHMARKING VISUAL PERCEPTION OF LLMS VIA ASCIIEVAL

We first assess model performance on text inputs, then propose rationale-assisted fine-tuning to enhance LLMs' recognition ability and discuss future directions.

## 5.1 PERFORMANCE OF LLMS

Performance of LLMs and proprietary models with only text inputs is shown in Fig. 2. Fig. 2(a) only presents a leaderboard with top-12 models. The full leaderboard is in Appendix H.

**Overall Performances**   All of the models in Fig 2(a) exceeds a random baseline (25%), confirming their fundamental competence on visual perception through text strings. However, a significant performance disparity exists between proprietary and open-source models. The former dominate the upper echelons of the leaderboard. The leading proprietary model, GPT-5, outperforming its open-source counterpart, DeepSeek-V3, by a substantial margin of 19.96%. Nevertheless, all models lags far behind the human upperbound (98.33%), reflecting the difficulty of our benchmark.

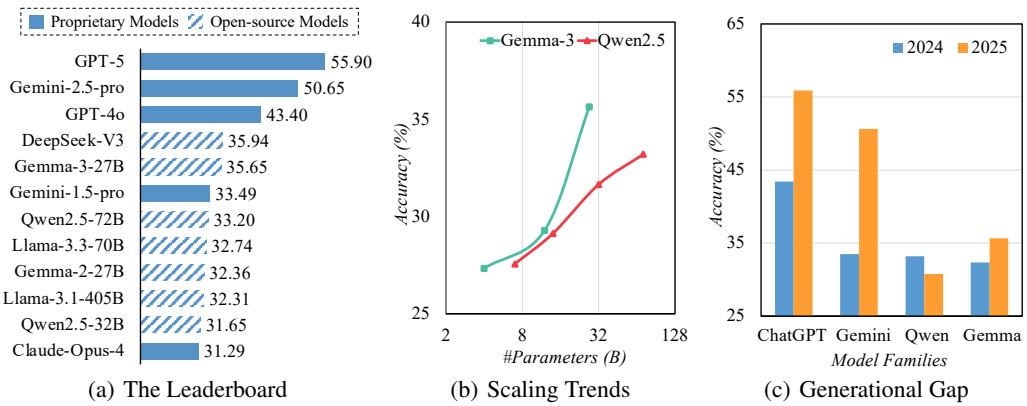

Figure 2: Macro-accuracy(%) of LLMs on ASCIIEVAL.

**Scaling Trends**   We plot the performance against the parameter count for representative models from the Gemma-3 and Qwen2.5 series in Fig 2(b). The results indicate clear scaling trends within each single-model series. However, this scaling law does not hold across different model series. Gemma-3 with only 27B parameters outperforms other competitors with more than 70B and even hundreds of billions of parameters. This underscores the potential of developing powerful lightweight models with strong visual perception abilities.

**Generational Gap**   Fig 2(c) compares the performances of models released in 2024 with their successors from 2025 across four model families. Proprietary models indicate substantial improvements across years, with accuracy gains exceeding 10%. In contrast, open-source models exhibit a tread of stagnation, widening the performance gap between proprietary and open-source models.

**Correlation Analysis**   ASCII art is not the only form of visual information embedded in text. Other representations, such as tabular data and code snippets with spatial significance, share a similar underlying requirement for this fundamental capability. To confirm this shared capability, we compared our benchmark against TableEval (Zhu et al., 2025b) and SGP-Bench (Qiu et al., 2025b), which assess LLMs on table question-answering and symbolic graphics understanding, respectively. The results show a strong positive correlation between performance on our dataset and these two benchmarks, with Pearson correlations of 0.78 and 0.85. While these findings suggest a shared fundamental skill, they also underscore the unique value of our benchmark. ASCIIEval isolates the core visual perception ability from other confounding factors such as complex reasoning, providing a more challenging and focused evaluation.

## 5.2   IMPROVING LLMs BY RATIONALE-ASSISTED FINE-TUNING

Our preliminary experiments revealed that fine-tuning LLMs on the ASCIITUNE by generating the choice given the multiple choice question with textual ASCII arts directly fails to yield improvements in their visual perception capabilities. Inspired by the outstanding performance of GPT-5 given the image input in Sec. 6, and the success of LLMs' reasoning ability by encouraging chain-of-thought, we propose rationale-assisted fine-tuning. This approach is designed to explicitly teach the model the underlying analytical process required for interpreting complex ASCII art, rather than merely exposing it to input-output pairs. It includes two primary stages as follows:

**Data Synthesis**   The cornerstone of our approach is the creation of a high-quality, rationale-annotated dataset. Recognizing the superior performance of state-of-the-art proprietary models, we employ GPT-5 given both $x_{\text{text}}$ and $x_{\text{img}}$ to synthesize the reasoning process in rich of the interpretation of local ASCII art features. 6309 instances are left after data verification.

**Rationale-assisted Fine-tuning**   We fine-tune the LLM on the synthesized dataset. For each instance, the model receives the original ASCII art $x_{text}$ as input. The target output is the concatenated string of the rationale and the oracle answer $y$. Further details are in the Appendix I.

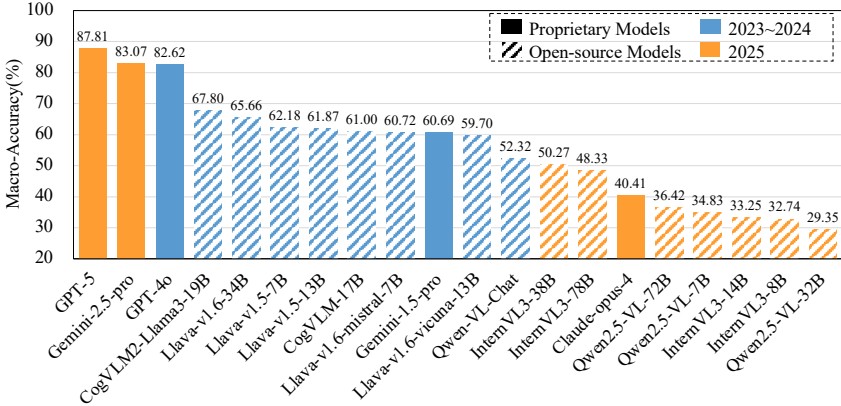

Figure 4: Macro-accuracy(%) of MLLMs on ASCIIEVAL.

Using Qwen3-8B as the backbone, we found that both zero-shot with thinking and fine-tuning on the ASCIITUNE failed to improve performance, achieving 27.21% and 26.23% respectively. In contrast, rationale-assisted fine-tuning significantly elevated the model's accuracy from its original 28.28% to 35.66%, a relative gain of 26.10%. This improvement propelled the model to fifth place on the leaderboard. Our method enabled this smaller model to outperform not only open-source models with a significantly larger number of parameters but also several proprietary models.

## 5.3 FUTURE DIRECTIONS

Although Rationale-Assisted Training significantly enhances model performance, we posit that this improvement does not fundamentally enhance LLMs' ability. Its success stems from a divide-and-conquer strategy. The rationale effectively deconstructs a complex ASCII art into a series of localized sub-strings with descriptions, assisting LLMs to perform compositional reasoning at the inference time by identifying and recombining fragments memorized during training. We hypothesize that the bottleneck lies in the tokenization process of LLMs, which is inherently unsuitable for preserving 2D spatial information. For example, the dog will be processed into 13 tokens as shown in Fig 3. Consecutive characters will be con-

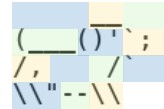

Figure 3: An illustration of the tokenized ASCII art. Each colored block represents a token.

catenated arbitrarily, which inevitably destroys the crucial vertical coherence of the art. Therefore, exploring alternative input representations is a vital furture direction.

## 6 BENCHMARKING AND ENHANCING MLLMS ON ASCIIEval

We evaluate models on image inputs, introduce two strategies and also discuss future directions. More analysis on MLLMs' sensitivity to minor character changes and fonts is in Appendix K and L.

## 6.1 PERFORMANCE OF MLLMS

**Overall Performance** Our evaluation reveals a clear performance hierarchy among contemporary MLLMs. At the apex of the leaderboard, proprietary models demonstrate superior capabilities, with GPT-5 achieving the highest accuracy of 87.81%, closely followed by Gemini-2.5-pro. The top-performing open-source model, CogVLM2, attains a respectable accuracy of 67.80% despite its relatively modest 19B parameter count. Nevertheless, a substantial performance gap persists between the two ecosystems. GPT-5 outperforms the leading open-source model by a significant margin of 20.01%, underscoring the current dominance of proprietary models on this visual perception task.

**Generational Gap** A longitudinal analysis comparing models released in 2023 to 2024 with their 2025 successors highlights a diverging trend in development. Proprietary models exhibit significant year-over-year improvement, indicating a rapid advancement in their ability to interpret the abstract,

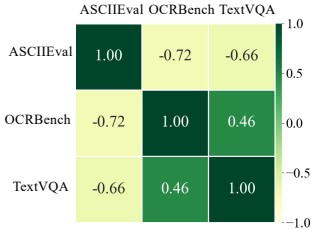

Figure 5: Pearson Correlations between multi-modal benchmarks.

Table 2: Macro-accuracy(%) of Qwen2.5-VL-7B on ASCIIEVAL by different approaches. The **best** and sub-optimal results are in bold and underlined.

**(a) Low-resolution prompting**

| Resolution | Accuracy |
|---|---|
| default | 34.83 |
| (1, 16) | **52.32** |
| (1, 32) | 47.65 |
| (1, 64) | 40.59 |
| (1, 128) | 38.81 |

**(b) Fine-tuning strategies**

| Method | Accuracy |
|---|---|
| zero-shot | 34.83 |
| full-parameter fine-tuning | **75.83** |
| LoRA *w.* Image Encoder | 75.48 |
| LoRA *w.* Text Backbone | 35.99 |
| LoRA *w.* Both | 74.23 |

symbolic nature of text strings. For instance, the Gemini family's accuracy surged from 60.69% to 82.62%. In contrast, the open-source models exhibit a marked decline in performance. Taking the Qwen-VL family as an example, the earlier model achieves 52.32% accuracy, whereas its successor with the same number of parameters only reached 34.83%. This regression suggests that the focus of open-source model development may be shifting away from core visual interpretation capabilities.

**Correlation Analysis**  By analyzing the outputs (see Appendix O), we found that open-source MLLMs with stronger OCR capabilities tend to "read" the characters while neglecting to "see" the emergent visual information they collectively form. We hypothesize that their performance decline on ASCIIEval stems from an overemphasis on OCR abilities. We analyze the correlation between open-source MLLMs' performance on ASCIIEval and OCR-centric benchmarks, including OCRBench (Liu et al., 2024b) and TextVQA (Singh et al., 2019). The results in Fig. 5 show a strong negative correlation. With this hypothesis in mind, we propose improving MLLM performance by deliberately "blurring" the input in Sec. 6.2, the success of which supports our hypothesis.

## 6.2 Improving MLLMs by Low-resolution Prompting and Fine-tuning

We explore two strategies to improve the performance of MLLMs.

**Low-resolution Prompting:**  We propose a test-time strategy by deliberately obscuring specific characters and compelling the model to percept global visual cues. We conduct experiments on Qwen2.5-VL-7B, which features the flexibility to accept a wide range of input resolutions. We set the minimum number of pixels to 1 and compared performance with a varying maximum number of pixels across the set $\{16, 32, 64, 128\}$. Results in Table 2(a) indicates a clear inverse correlation, with the lowest resolution yielding the highest accuracy. The model achieved 52.32% accuracy at the lowest resolution setting, outperforms the default baseline by 17.49%. This finding challenges the common assumption that higher resolutions lead to better performance, suggesting that intentionally downscaling images to blur fine details is necessary in certain scenarios.

**Supervised Fine-tuning:**  We investigate whether supervised fine-tuning can enhance the MLLM's capability for text-based visual perception. Using ASCIITUNE, the model was provided with the ASCII art image and trained to generate the correct textual answer. We train Qwen2.5-VL-7B with different fine-tuning strategies, including full-parameter fine-tuning, and parameter-efficient fine-tuning by low-rank adaptation (LoRA) on QKV matrices from different model components. As shown in Table 2(b), the results highlight that fine-tuning the vision backbone plays the critical factor for performance improvements. Applying LoRA solely on the visual backbone achieves 75.48%, nearly matching the full-parameter approach. Ultimately, this approach boosts Qwen2.5-VL-7B to the 4th place on the leaderboard, closely following strong proprietary models.

## 6.3 Future Directions

Our benchmark, ASCIIEval, highlights a critical but overlooked dimension of visual intelligence: holistic visual understanding. It reveals a fundamental trade-off, showing that an overemphasis on fine-grained text recognition can come at the expense of a model's ability to perceive collective visual information. While our proposed methods including low-resolution prompting and supervised fine-tuning, efficiently improve ASCII art performance without compromising the base model's core

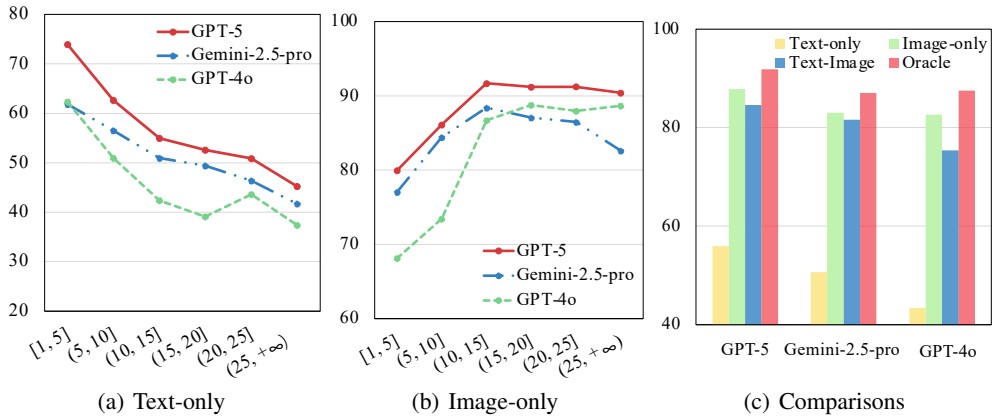

Figure 6: Micro-accuracy (%) of models on ASCII art with different numbers of characters.

capabilities, they are merely pos-hoc solutions. Developing models that can intrinsically balance these competing skills is crucial for achieving the robust, state-of-the-art performance observed in leading proprietary models and for complicated real applications.

## 7 THE ABSENCE OF INTER-MODAL SYNERGY IN MLLMS

To investigate how the complexity of ASCII art influences model performance, we analyzed test samples partitioned into six subsets based on their line count, with the results shown in Fig.6. Models under the text-only setting demonstrate a proficiency in recognizing shorter ASCII art, where significant features are often densely packed within consecutive characters. For instance, the string "() '`;" concisely captures key features of a dog (Fig.1), suggesting that LLMs excel at associating concepts with these dense, local character patterns. However, as the size increases, these localized features become diluted, demanding a stronger 2D perceptual ability that text-only models inherently lack. Conversely, models given the image inputs are more adept at interpreting larger ASCII art. This is because smaller, more abstract pieces bear little resemblance to their training data, whereas larger creations are structurally similar to real images and posters they were trained on, sharing comparable outlines and luminance contrasts, as seen with the Spiderman in Fig. 1.

A key finding across our experiments is a consistent performance hierarchy: Image-only > Text-Image > Text-only. We introduce "oracle" in Fig. 6(c) as a performance ceiling, which deems a prediction correct if the model succeeds with either modality alone. Our results reveals that: the inclusion of textual information alongside the image consistently impairs model performance, rather than enabling it to approach this upper bound. Specifically, all models exhibited a performance drop in the Text-Image setting compared to the Image-only baseline, with this degradation reaching up to 12.23%. This exposes a fundamental weakness that, instead of effectively leveraging the complementarity and consistency between visual and textual data, current MLLMs appear to be confounded by the concurrent inputs, leading to a higher error rate.

**Future Directions:** The demonstrated failure of modality fusion presents a critical area for future research. Future research should therefore prioritize elucidating the internal mechanisms of modal conflict while developing architectures capable of dynamic fusion. Achieving this is a crucial step toward building robust models that flexibly synthesize all available information for a more holistic and accurate understanding.

## 8 CONCLUSION

In this work, we focus on analyzing and eliciting models' visual perception ability in text strings via ASCII arts. We introduce the ASCII art recognition problem, which task models to recognize the concepts depicted by the art conveyed through different modalities. We constructed both test and training data, and conducted comprehensive evaluations with dozens of LLMs and MLLMs

followed by multiple enhancement approaches. Results pinpoint that our benchmark serves as a more challenging for benchmarking LLMs' visual perception ability and MLLMs' holistic visual understanding ability. It also reveal a lack of effective fusion techniques for semantic-equivalent information across different modalities, highlighting multiple future directions.

## ACKNOWLEDGEMENT

This work was supported by New Generation Artificial Intelligence-National Science and Technology Major Project (2025ZD0124104) in collaboration with Shanghai Artificial Intelligence Laboratory. It is also supported by Prof. Yang You, whose research group is being sponsored by NUS startup grant (Presidential Young Professorship), Singapore MOE Tier-1 grant, ByteDance grant, NUS ARTIC grant, Apple grant, Alibaba grant and Adobe gift.

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

## A    THE USE OF LARGE LANGUAGE MODELS

Large language models, including Gemini-2.5-pro and Gemini-2.5-Flash, were adopted to polish the writing in this work. All of the models' outputs are verified and further modified by the authors. LLMs did not participate in any other stages of the research process.

## B    DATA LICENSE

We express our gratitude to ASCII artists from online galleries whose fantastic creations underpin our research. In order to assess the visual perception abilities of models, we made slight modifications to the original ASCII art for the test set ASCIIEVAL, to avoid information leakage through text hints. Meanwhile, we retained the original ASCII art and the URL to the data source. We follows the term of use guidelines from the original websites [2] and datasets [3]. Data will be released and licensed under CC BY NC 4.0, which permits only non-commercial use and is intended exclusively for research purposes.

## C    FUTURE DIRECTIONS

Based on the results and analysis, we discuss more future directions as follows:

**Constructing high-quality training data automatically.** We randomly selected 100 samples from ASCIITUNE for the quality check and the human annotator achieved only 70% accuracy. This indicates that ASCIITUNE is much noisier than ASCIIEVAL (98.33%), pointing out the importance of collecting more training data with higher quality. On the one hand, utilizing the ASCII art synthesis tools to convert image datasets into ASCII art can be considered to enlarge the size of the training data, under the awareness of the style differences between the converted ones and the ones created by artists. On the other hand, more strict filtering strategies should be incorporated, such as verifying the validity of ASCII art with strong MLLMs under the Image-only setting.

**Improving the model architecture.** All of the tested LLMs and MLLMs show the inability to recognize information that can be fully represented in text. One potential reason is the lack of exposure to this type of data. It may be also a result of the structural limitation of current models. As for human beings, we perceive text from the aspects of character sequences and their visual shapes at the same time, while these two aspects are conventionally distinguished into two modalities when being processed by neural models. More flexible processing techniques and architecture among modalities should not only benefit the models' visual perception ability in text strings, but also make the model closer to human beings with more efficient information processing abilities.

**Incorporating more complicated scenarios.** Currently, we only considered the basic type of ASCII art made up of 95 printable fixed-width ASCII characters. Nevertheless, there also exist more fascinating ASCII arts, such as color ASCII art, 3D ASCII art, animated ASCII art, etc. These different kinds of ASCII art are also valuable for understanding LLMs designed for video understanding (He et al., 2024a) and 3D modeling (Hong et al., 2023).

## D    PROMPT TEMPLATE

We adopted the following three prompt templates for different input modes:

*Prompt Template for Text-only Input*

```
Please answer the multi-choice question based on the given ASCII
art:

[ASCII ART]
ascii_art
```

---

[2] https://asciiart.website/, https://ascii.co.uk/art
[3] https://huggingface.co/datasets/apehex/ascii-art

```
[Question]
What is depicted in the above ASCII art?  {choices}

Answer with the option's letter from the given choices directly.
```

*Prompt Template for Image-only Input*

```
Please answer the multi-choice question based on the given ASCII
art image.

[ASCII ART]
<image>

[Question]
What is depicted in the above ASCII art?  {choices}

Answer with the option's letter from the given choices directly.
```

*Prompt Template for Image-text Input*

```
Please answer the multi-choice question based on the given ASCII
art in both image and text formats.

[ASCII ART Image]
<image>

[ASCII ART Text]
ascii_art

[Question]
What is depicted in the above ASCII art?  {choices}

Answer with the option's letter from the given choices directly.
```

All of the models except Qwen-VL are evaluated based on these prompt templates with minor modifications to adapt to their default settings, especially for the position of the image.

Qwen-VL is more sensitive to prompt templates according our experiments. Therefore, we adapted the above templates into Qwen-VL's original format, which is "Context: ... Question: ... Answer:".

## E    DATA COLLECTION FOR ASCIITUNE

To further elicit models' visual perception ability, the creation of a training set is essential. An intuitive solution is to leverage previous works on ASCII art synthesis (Xu et al., 2016; 2010) by converting existing image datasets, such as ImageNet (Deng et al., 2009). A public dataset [4] indicates that after automatic tone-based synthesis, approximately 85% data samples are filtered out due to poor quality. Furthermore, existing data conversion tools are inadequate for structure-based ASCII art, which accounts for 94% of the data according to annotators' labels in ASCIIEVAL. Artists also frequently combine both tone-based and structure-based features in a single artifact.

Therefore, we chose to collect the training set in a manner similar to ASCIIEVAL instead of relying on automatic conversion. Data sources include ASCII arts from another less organized website [5], and the crawled content was extracted into individual ASCII art pieces based on rules derived from observations. We also included the unrecognized ASCII art that was withdrawn during the construction of ASCIIEVAL. The normalized ASCII art is discarded if recognized as repetitive with samples in ASCIIEVAL or among each other.

---

[4] https://huggingface.co/datasets/mrzjy/ascii_art_generation_140k
[5] https://ascii.co.uk/art

Table 3: The number of samples under each category.

| Classes | Groups |
|---|---|
| animals & natural (1,122) | animal (870), plant (130), nature (122) |
| objects (777) | object (451), electronics (192), clothing (81), furniture (53) |
| smileys & people (644) | role (199), character (195), body (146), occupation (68), people (36) |
| activities (473) | event (207), sport (126), activity (84), instrument (35), monument (21) |
| travel & places (406) | transportation (123), building (123), places (30) |
| food & drink (66) | food (66) |
| symbols (38) | logo (27), astrology (11) |

Table 4: Statistics of token length by different tokenizers.

| | ASCIIEVAL | | | ASCIITUNE | | |
|---|---|---|---|---|---|---|
| | Min | Max | Avg | Min | Max | Avg |
| Llama-3 Tokenizer | 71 | 2,192 | 262.72 | 69 | 3,673 | 215.10 |
| Mistral-v0.1 Tokenizer | 85 | 2,890 | 332.91 | 83 | 4,294 | 267.93 |
| Qwen-2 Tokenizer | 80 | 2,833 | 278.17 | 78 | 3,996 | 273.40 |

Due to the large amount of data with diverse concepts, carefully categorizing data for high-quality distractors is unfeasible. Instead, we prompted Llama-3-70B-Instruct to generate negative choices given the ground truth concept and utilized the Perspective API to filter out unsafe samples based on the concatenation of candidate choices. Samples with scores less than 0.2 across all six dimensions, i.e., toxicity, severe toxicity, identity attack, insult, profanity and threat, are retained.

## F  DATA ANALYSIS AND STATISTICS

During the data filtering process, we recognized that some of the ASCII art have multiple interpretations, which can be summarized into two types:

○ The ASCII art itself, as a kind of art form, is abstract and ambiguous. For instance, certain depictions of cats might resemble rats. Regarding these cases, we asked human annotators to remove such unrecognizable and ambiguous art.

○ The ASCII art is rich in content, potentially allowing two interpretations from different aspects. For example, the third ASCII art in Fig. 15, can be interpreted as a beach scene, coconut tree, sunset, etc. Most of the ASCII art in ASCIIEVAL only contains a single object, and we also tried to remove such ambiguities by carefully designing and adjusting the classification criterion. Ultimately, there are only less than 1.67% ambiguous cases in ASCIIEVAL, leading to the imperfect performance of human annotators.

Finally, the number of samples and the hierarchical relationship between classes and groups of ASCIIEVAL illustrated are shown in Table 3.

The token length of samples under the Text-only mode tokenized by three representative tokenizers is in Table 4. The ASCII art data used in our experiments respects the context length limitation of nowadays models.

## G  DETAILS ABOUT EVALUATED MODELS

For open-source instructed models, we experiment with the following LLMs and MLLMs:

**LLMs.**  **Llama** (Touvron et al., 2023) contains three collections of generative models with different sizes, including Llama-2, Llama-3, Llama-3.1, and Llama-3.3; **Qwen** (Bai et al., 2023a; Team, 2024b; Yang et al., 2025) is another group of models with instructed verions, including Qwen, Qwen1.5, Qwen2, Qwen2.5 and Qwen3 series; **Mistral** (Jiang et al., 2024a) includes different versions of instruction fine-tuned models, i.e., Mistral-7B-Instruct-v0.1, v0.2 and v0.3. Besides, Mixtral-8x7B-Instruct-v0.1 and Mixtral-8x22B-Instruct-v0.1 which are pre-trained generative Sparse Mixture of Experts are also compared; **Gemma** (Team, 2024a; Team et al., 2025) is a fam-

ily of lightweight text-to-text models with instruction-tuned variants. We considered Gemma-2 and Gemma-3 series; **DeepSeek** Liu et al. (2024a) is a series of open-source large language models, with DeepSeek-V3 being a notable model in this family.

**MLLMs.** **Llava** (Liu et al., 2023) augmented a pre-trained LLM with a pre-trained vision encoder. The vision model's representations are projected into the LLM's representation space with a projection layer, and it is frozen during instruction tuning while the projector and the backbone LLM are updated; **CogVLM** (Wang et al., 2024b) aims at retaining the original capabilities of the LLM while adding visual understanding abilities. Representations from the pre-trained vision transformer encoder are passed through an MLP adapter as the input, and a group of trainable visual expert modules in the attention and FFN layers are introduced into the LLM. All of the parameters except the ones from the original LLM are tuned; **Qwen-VL** (Bai et al., 2023b) proposed a position-aware vision-language adapter for compressing image features. The model is trained through three stages, i.e., pre-training, multi-task pre-training and supervised fine-tuning; **Qwen2.5-VL** (Bai et al., 2025) introduce dynamic resolution processing ad excelling in omni-document parsing; **InternVL3** consolidates language pre-training and multi-modal alignment training into a unified pre-training stage with interleaving multi-modal data.

For proprietary models, the specific versions we evaluated are GPT-4o-20240806 (OpenAI, 2023), GPT-5-20250807, Claude-opus-4-20250514, Gemini-1.5-pro (Reid et al., 2024) and Gemini-2.5-pro Comanici et al. (2025).

## H  ASCIIEVAL LEADERBOARD

We summarize the above models performance on ASCIIEVAL given different input modalities in Table 5. The statistics used for calculating correlations in Sec. 5.1 and Sec. 6.1 were collected by extracting scores of the overlapped models covered both in Table 5 and corresponding leaderboards [6].

## I  DATA SYNTHESIS AND TRAINING DETAILS

Recognizing the superior performance of state-of-the-art proprietary models, we devised a multi-step data synthesis pipeline:

○ *Data Curation:* We first employ a high-performing open-source model to filter the ASCIITune. This initial pass serves to remove low-quality or ambiguous samples, yielding 8925 samples.

○ *Rationale Generation:* For each curated data point, we provide the teacher model, GPT-5, with both $x_{\text{text}}$ and $x_{\text{img}}$. The model is prompted to first generate a detailed analytical process, i.e. rationale, that describes its the reasoning process for recognition in rich of the interpretation of local ASCII art features, with the answer at the end of the output.

○ *Fidelity Verification:* Only 6309 instances where GPT-5's final answer is correct are retained.

An example of the output distilled from GPT-5 is shown in Fig. 7. Different colors marks the corresponding string in output text and the ASCII image. The output analysis explains the details of the model's perception process reasonably, but also contains minor errors. ")/_" is a piece of hallucinated text string which not included in the original ASCII art. Employing more rigorous filtering strategies to remove such mistakes for high-quality data collection will be considered in the future.

The LLMs in Sec. 5.2 is finetuned on the distilled data for 2 epoch on full parameters, with batch size equaling 16 and learning rate equaling 2e-5. The MLLMs trained in Sec. 6.2 adopted the same batch size and learnin rate. We did fine-tuning with full parameters for 1 epoch and with LoRA for 2 epochs.

Table 5: ASCIIEVAL Leaderboard. The scores are macro accuracy (%) averaged among different concepts. Average refers to the mean among the three input settings if available. All of the models are "instruct" or "chat" versions. The **best** and sub-optimal results in each group of models are in bold and underlined. The $\pm values$ mark the *95% confidence interval* for concept-level ASCII art recognition.

| Model | Text-only | Image-only | Text-Image | Average |
|---|---|---|---|---|
| *Proprietary Models* | | | | |
| GPT-5 | **55.90**$_{\pm 8.96}$ | **87.81**$_{\pm 5.54}$ | **86.40**$_{\pm 5.57}$ | **76.70** |
| GPT-4o | 43.40$_{\pm 8.40}$ | 82.62$_{\pm 5.31}$ | 75.41$_{\pm 7.77}$ | 67.14 |
| Gemini-2.5-pro | 50.65$_{\pm 7.22}$ | 83.07$_{\pm 6.13}$ | 81.64$_{\pm 6.17}$ | 71.79 |
| Gemini-1.5-pro | 33.49$_{\pm 8.76}$ | 60.69$_{\pm 6.50}$ | 58.33$_{\pm 7.68}$ | 50.84 |
| Claude-opus-4 | 31.29$_{\pm 7.04}$ | 40.41$_{\pm 7.03}$ | 36.68$_{\pm 12.16}$ | 36.13 |
| *Open-source LLMs* | | | | |
| DeepSeek-V3 | **35.94**$_{\pm 6.74}$ | - | - | **35.94** |
| Qwen3-8B | 28.28$_{\pm 7.13}$ | - | - | 28.28 |
| Qwen3-14B | 30.79$_{\pm 6.21}$ | - | - | 30.79 |
| Qwen3-32B | 30.18$_{\pm 7.64}$ | - | - | 30.18 |
| Qwen2.5-7B | 27.57$_{\pm 4.77}$ | - | - | 27.57 |
| Qwen2.5-14B | 29.14$_{\pm 7.17}$ | - | - | 29.14 |
| Qwen2.5-32B | 31.65$_{\pm 6.74}$ | - | - | 31.65 |
| Qwen2.5-72B | 33.20$_{\pm 5.05}$ | - | - | 33.20 |
| Qwen2-7B | 27.71$_{\pm 6.69}$ | - | - | 27.71 |
| Qwen2-72B | 30.73$_{\pm 6.54}$ | - | - | 30.73 |
| Qwen1.5-7B | 26.71$_{\pm 7.77}$ | - | - | 26.71 |
| Qwen1.5-110B | 30.28$_{\pm 7.20}$ | - | - | 30.28 |
| Qwen-7B | 23.30$_{\pm 6.42}$ | - | - | 23.30 |
| Gemma-3-4B | 27.34$_{\pm 7.39}$ | - | - | 27.34 |
| Gemma-3-12B | 29.29$_{\pm 8.39}$ | - | - | 29.29 |
| Gemma-3-27B | 35.65$_{\pm 7.66}$ | - | - | 35.65 |
| Gemma-2-9B | 30.50$_{\pm 7.37}$ | - | - | 30.50 |
| Gemma-2-27B | 32.36$_{\pm 6.49}$ | - | - | 32.36 |
| Llama-3.3-70B | 32.74$_{\pm 5.17}$ | - | - | 32.74 |
| Llama-3.1-8B | 27.22$_{\pm 6.66}$ | - | - | 27.22 |
| Llama-3.1-70B | 31.27$_{\pm 5.44}$ | - | - | 31.27 |
| Llama-3.1-405B | 32.31$_{\pm 7.43}$ | - | - | 32.31 |
| Llama-3-8B | 28.71$_{\pm 8.00}$ | - | - | 28.71 |
| Llama-3-70B | 30.42$_{\pm 4.46}$ | - | - | 30.42 |
| Llama-2-7B | 24.59$_{\pm 7.61}$ | - | - | 24.59 |
| Llama-2-13B | 25.93$_{\pm 7.59}$ | - | - | 25.93 |
| Llama-2-70B | 28.08$_{\pm 6.93}$ | - | - | 28.08 |
| Mistral-7B-v0.1 | 26.88$_{\pm 7.64}$ | - | - | 26.88 |
| Mistral-7B-v0.2 | 26.28$_{\pm 7.44}$ | - | - | 26.28 |
| Mistral-7B-v0.3 | 25.57$_{\pm 7.79}$ | - | - | 25.57 |
| Mixtral-8x7B-v0.1 | 25.31$_{\pm 6.93}$ | - | - | 25.31 |
| Mixtral-8x22B-v0.1 | 28.20$_{\pm 6.51}$ | - | - | 28.20 |
| *Open-source MLLMs* | | | | |
| Qwen2.5-VL-7B | 25.05$_{\pm 7.21}$ | 34.83$_{\pm 7.90}$ | 37.01$_{\pm 9.31}$ | 32.30 |
| Qwen2.5-VL-32B | 29.82$_{\pm 8.42}$ | 29.35$_{\pm 7.22}$ | 32.07$_{\pm 8.79}$ | 30.41 |
| Qwen2.5-VL-72B | **34.20**$_{\pm 7.62}$ | 36.42$_{\pm 7.87}$ | 37.82$_{\pm 7.66}$ | 36.15 |
| Qwen-VL | 24.79$_{\pm 7.58}$ | 52.32$_{\pm 8.06}$ | 40.09$_{\pm 7.84}$ | 39.07 |
| InternVL3-8B | 27.30$_{\pm 7.39}$ | 32.74$_{\pm 6.04}$ | 33.58$_{\pm 6.03}$ | 31.21 |
| InternVL3-14B | 25.91$_{\pm 6.76}$ | 33.25$_{\pm 7.88}$ | 31.50$_{\pm 8.61}$ | 30.22 |
| InternVL3-38B | 32.10$_{\pm 7.69}$ | 50.27$_{\pm 7.68}$ | 47.28$_{\pm 8.60}$ | 43.22 |
| InternVL3-78B | 33.55$_{\pm 7.86}$ | 48.33$_{\pm 8.96}$ | 48.54$_{\pm 9.20}$ | 43.37 |
| CogVLM2-Llama3-19B | 24.73$_{\pm 6.88}$ | **67.80**$_{\pm 6.51}$ | **66.68**$_{\pm 6.77}$ | **53.07** |
| CogVLM-17B | 21.25$_{\pm 6.97}$ | 61.00$_{\pm 6.81}$ | 57.58$_{\pm 6.79}$ | 46.61 |
| Llava-v1.6-mistral-7B | 25.89$_{\pm 5.63}$ | 60.72$_{\pm 8.03}$ | 59.02$_{\pm 8.03}$ | 48.54 |
| Llava-v1.6-vicuna-13B | 26.03$_{\pm 6.90}$ | 59.70$_{\pm 7.63}$ | 56.55$_{\pm 7.65}$ | 47.43 |
| Llava-v1.5-7B | 24.66$_{\pm 6.64}$ | 62.18$_{\pm 7.43}$ | 61.52$_{\pm 8.23}$ | 49.45 |
| Llava-v1.5-13B | 26.00$_{\pm 5.94}$ | 61.87$_{\pm 7.42}$ | 60.70$_{\pm 7.02}$ | 49.52 |
| Llava-v1.6-34B | 28.62$_{\pm 5.40}$ | 65.66$_{\pm 7.43}$ | 61.33$_{\pm 7.84}$ | 51.87 |

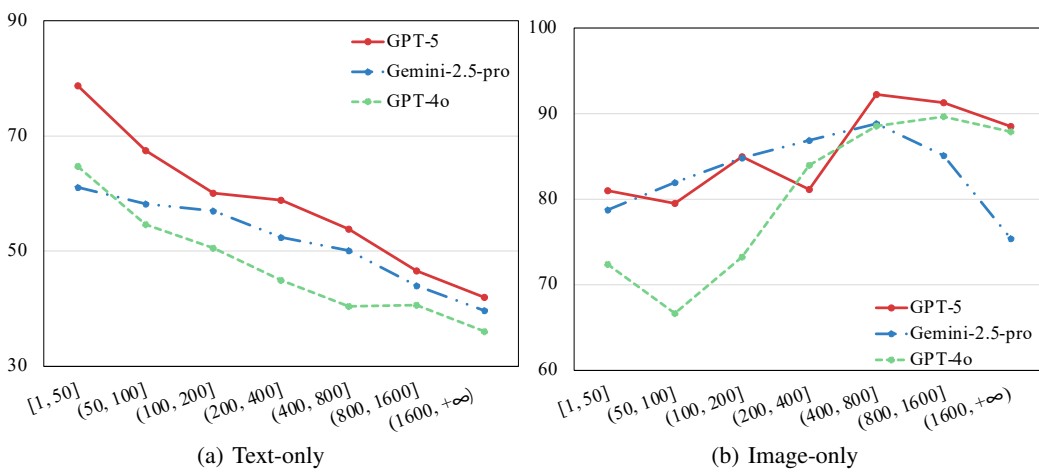

Figure 7: An example of distilled data for rationale-assisted fine-tuning.

Table 6: The number of samples with ASCII arts divided by different characteristics.

| #Characters | [1, 50] | (50,100] | (100, 200] | (200, 400] | (400, 800] | (800, 1600] | (1600, +∞) |
|---|---|---|---|---|---|---|---|
| #Samples | 221 | 366 | 546 | 710 | 760 | 618 | 305 |

| #Lines | [1,5] | (5, 10] | (10, 15] | (15, 20] | (20, 25] | (25,+∞) | - |
|---|---|---|---|---|---|---|---|
| #Samples | 414 | 854 | 699 | 534 | 399 | 626 | - |

(a) Text-only

(b) Image-only

Figure 8: Micro accuracy(%) of models on recognizing ASCII arts with different numbers of characters.

## J ANALYSIS ON SAMPLES UNDER DIFFERENT ASCII ART SIZES

Based on the length characteristics of different ASCII art, we divided the test set into various subsets, as shown in Table 6.

The performances of models on testing samples grouped by the number of lines contained in the ASCII art are shown in Fig. 8. The trends are similar to those grouped by the number of characters in Sec 7, i.e., models favor smaller ASCII art under the Text-only setting while they prefer larger ASCII art under the Image-only setting. Besides, when an ASCII art exceeds 800 characters, the model's performance tends to plateau or even degrade, underscoring that recognizing large-scale ASCII art also remains challenging for MLLMs.

## K SENSITIVITY TO MINOR CHARACTER CHANGES

We randomly removed tokens (other than spaces, "\n" and "\t") from ASCII art and manually checked if the result remained recognizable. Two representative examples are illustrated in Fig. 9. In both cases, the ASCII art remains recognizable when only few characters are removed. However, the first ASCII art becomes progressively indistinguishable as more characters are missing. Meanwhile,

---

[6]https://github.com/wenge-research/TableEval, https://sgp-bench.github.io/,https://huggingface.co/spaces/opencompass/open_vlm_leaderboard

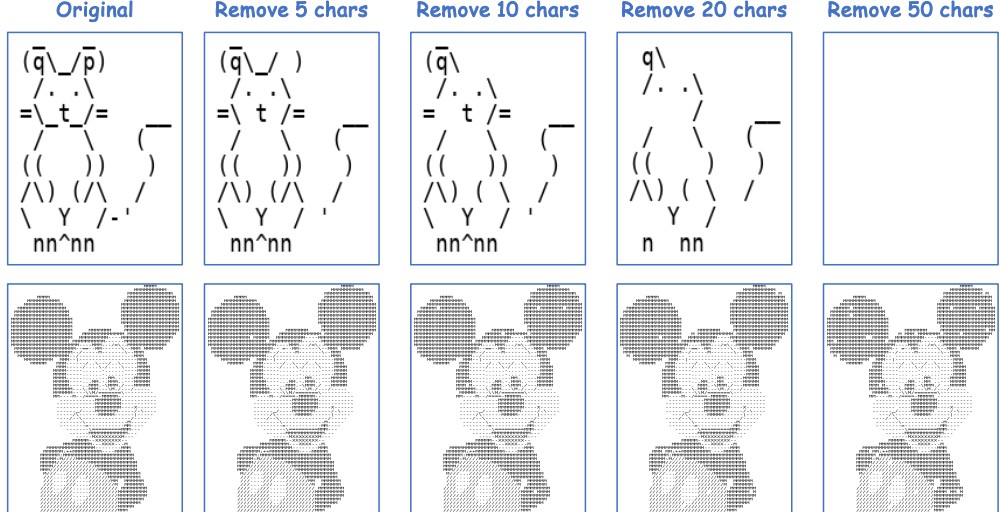

Figure 9: An illustration of removing characters in the ASCII art. "chars" is short for "characters".

Table 7: The micro-accuracy (%) at different perturbation ratios. "PR" is short for "Perturbation Ratio".

| PR | Human | Text-only | Image-only | Text-Image |
|---|---|---|---|---|
| 1% | 99 | 94 | 96 | 96 |
| 5% | 99 | 95 | 91 | 93 |
| 10% | 97 | 91 | 93 | 92 |
| 20% | 94 | 84 | 87 | 83 |

the second one just gradually has some additional noise and remains recognizable. This suggests that as the number of characters increases, the importance of each character diminishes as it carries less visual information.

We did more quantitative analysis by sampling 100 cases from ASCIIEVAL, among which Llava-v1.6-34B provided correct answers under all three test settings. Next, we randomly replaced 1%, 5%, 10%, and 20% of tokens (other than spaces, "\n" and "\t") in the original ASCII art with spaces.

The computed micro-accuracy of Llava-v1.6-34B under different test settings, as well as the human upper bound, are shown in Table 7. Changing the characters in ASCII art will make the recognition task more challenging both for humans and the model, while Human is relatively more robust than Llava-v1.6-34B under different settings.

## L  SENSITIVITY WITH DIFFERENT FONTS

In this work, we only considered the traditional ASCII art composed of 95 printable fixed-width ASCII characters. The semantic meaning remains unchanged as long as it is displayed with a fixed-width font. In addition to the "DejaVu Sans Mono" font used in this work, examples of the same ASCII art rendered with 4 different fonts are shown in Fig. 10. All of the dogs are recognizable, with only minor differences. In other words, the multiple-choice questions for ASCII art recognition in ASCIIEVAL remain valid, regardless of the specific fixed-width font used.

Although humans have no difficulty recognizing ASCII art rendered with different fonts, this raises the question of whether MLLMs are sensitive to these variations and show a preference to a specific fixed-width font. We take Llava-v1.6-34B as an example and evaluated its performance on ASCII art under both Image-only and Text-Image settings where the images are rendered using 5 different fonts mentioned in Fig. 10. It should be noted that the textual ASCII art is unaffected by font

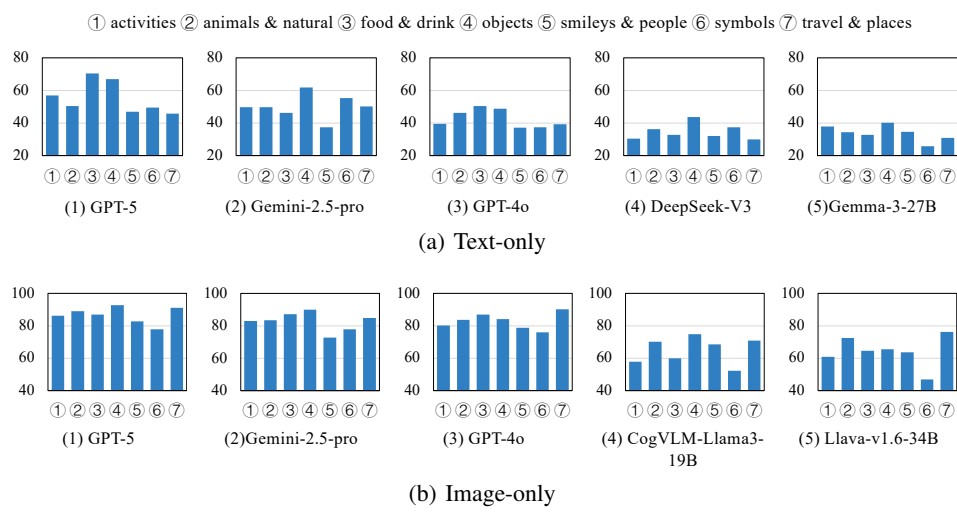

**DejaVu Sans Mono**   **Cascadia Code**   **Comic Mono**   **Courier**   **Fantasque Sans**

Figure 10: An illustration of an ASCII art displayed in different fixed-width fonts.

Table 8: Macro-accuracy(%) of Llava-v1.6-34B under Image-only and Text-Image setting with ASCII art rendered by different fix-width fonts.

| Mode | DejaVu Sans Mono | Cascadia Code | Comic Mono | Courier | Fantasque Sans |
|------|------------------|---------------|------------|---------|----------------|
| **Image-only** | 65.66 | 63.41 | 66.68 | 63.84 | 66.73 |
| **Text-Image** | 61.33 | 59.85 | 62.11 | 59.89 | 64.04 |

① activities ② animals & natural ③ food & drink ④ objects ⑤ smileys & people ⑥ symbols ⑦ travel & places

(a) Text-only

(b) Image-only

Figure 11: Macro-accuracy (%) of models on recognizing ASCII arts under different classes.

variations, and Llava-v1.6-34B's performance under the Text-only setting is identical to the result in Table 5.

According to the results in Table 8, MLLMs do face challenges in performing robustly among different text fonts in ASCII art recognition and the performance varies. Nevertheless, its best performance in this table with 66.73% and 64.04% still lags far behind that of GPT-4o with 83.69% and 76.52% under both settings respectively. Moreover, the accuracy under the Text-Image setting is consistently lower than that under the Image-only setting. These observations are same as the results in Sec. 7.

On the one hand, how to reduce this sensitivity and improve the MLLMs' robustness is important and worth further exploration. On the other hand, changing the fonts in rendered ASCII art can potentially a useful data augmentation technique for boosting MLLMs' performance on ASCIIEVAL.

# M  HOW DO MODELS PERFORM ON DIFFERENT CATEGORIES?

Models' performances across the 7 different classes are shown in Fig. 11. Models given text input perform better at recognizing ASCII arts belonging to the "objects" class. Models under Image-only mode show consistent improvement in recognizing "travel & places" over Text-only mode compared to other classes relatively. Moreover, all models struggle with ASCII art referring to "symbols", which comprise different logos and astrology symbols. MLLMs actually perform quite well at recognizing well-known logos, such as Apple and Linux, where GPT-5 achieves 100% macro-accuracy

| Models | [0,0.1] | (0.1,0.2] | (0.2,0.3] | (0.3,0.4] | (0.4,0.5] |
|--------|---------|-----------|-----------|-----------|-----------|
| DeepSeek-V3 | 7 | 220 | 834 | 1249 | 908 |
| Gemma-3-27B | 2 | 171 | 812 | 1158 | 913 |
| Qwen2.5-72B | 10 | 242 | 923 | 1286 | 839 |

| Models | (0.5,0.6] | (0.6,0.7] | (0.7,0.8] | (0.8,0.9] | (0.9,1.0] |
|--------|-----------|-----------|-----------|-----------|-----------|
| DeepSeek-V3 | 275 | 27 | 6 | 0 | 0 |
| Gemma-3-27B | 361 | 87 | 15 | 6 | 1 |
| Qwen2.5-72B | 191 | 26 | 6 | 3 | 0 |

Table 9: Number of samples in different compression ratio buckets.

and CogVLM2-Llama3-Chat-19B gets 91.16%. However, their performance drops dramatically on relatively niche astrology symbols. Nevertheless, it is simple for both LLMs and MLLMs to answer the question "Can you show me some astrology symbols?". Existing models tend to use rare Unicode characters or emojis to explain the symbols, but cannot understand the visual semantics embedded in those symbols flexibly.

The models' performance under different groups is shown in Fig. 12. Overall, the performance of models under Image-only mode is more balanced across different categories, except for the drops in "astrology" and "character". Meanwhile, accuracy given images fluctuates among different groups, with "electronics", "monumnet" and "object" topping the rank.

## N  DOES TOKENIZER COMPRESSION RATIO AFFECT MODEL PERFORMANCE?

This section explores whether the compression ratio of a model's tokenizer correlates with its accuracy on ASCII art recognition. We define the compression ratio for each ASCII art as the number of tokens generated by a model's tokenizer divided by the total number of characters in the string. A lower ratio indicates higher compression. To analyze this relationship, we group the test samples into ten buckets based on their compression ratio for top-3 open-source LLMs in Fig 2(a).

The results presented in Figure 13 reveals no single, consistent trend across all evaluated models. While two models, DeepSeek-V3 and Gemma-3-27B, exhibit a U-shaped performance curve, this pattern is not universal. Qwen2.5-72B does not follow this trend. Furthermore, according to Table 9, the apparent high performance at the extreme ends of the ratio spectrum is based on a sparse data set, making the results less reliable and prone to variance.

Given the lack of a unified performance trend across all models and the sparse data distribution at the extremes of the compression ratio, we conclude that there is no clear correlation between the tokenizer compression ratio and model performance on this task.

## O  CASE STUDY ON MODELS WITH A STRONG OCR ABILITY

MLLMs with strong OCR capabilities often struggle with ASCII art as shown in Sec. 6. They tend to prioritize recognizing individual characters over perceiving the holistic image formed by them. An example is shown in Fig. 14. The model Qwen2.5-VL-72B incorrectly classified an ASCII art of a mouse as a butterfly.

To understand this failure, we prompted the model with the question, "What is depicted in the above ASCII art?". Its response reveals a complete focus on character-level details instead of the overall shape:

> *The image appears to be a stylized or abstract representation of a mathematical or logical expression. It includes symbols and characters that resemble logical operators and variables, such as "q", "p", "t", and "nn". The symbols are arranged in a way that suggests a logical or mathematical formula, but the exact meaning or context is not clear without additional information. The use of slashes, dots, and other symbols might be part of a specific notation or a creative interpretation of logical expressions.*

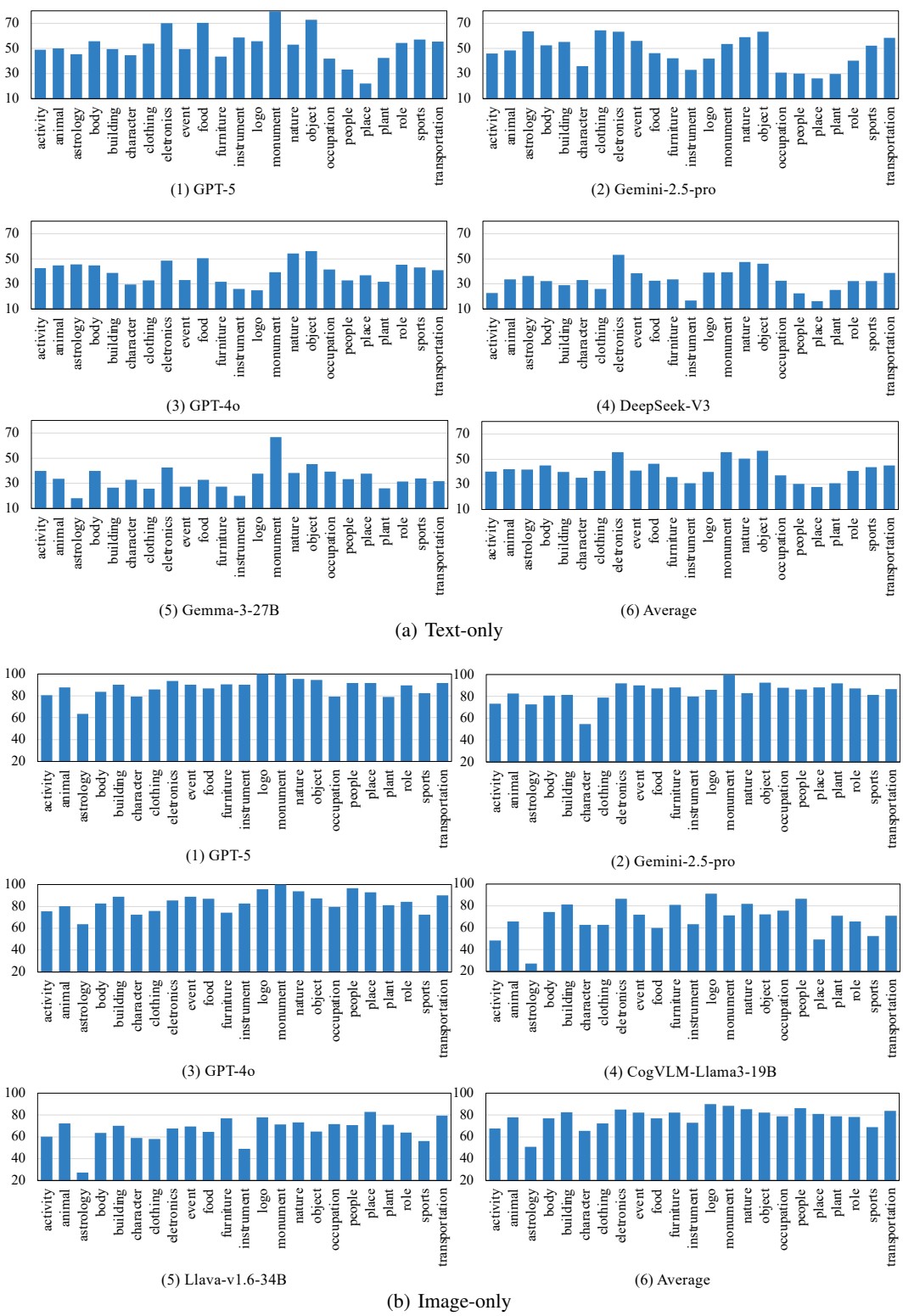

Figure 12: Micro accuracy(%) of models on recognizing ASCII arts in different groups. Average is calculated as the mean of the top 5 models.

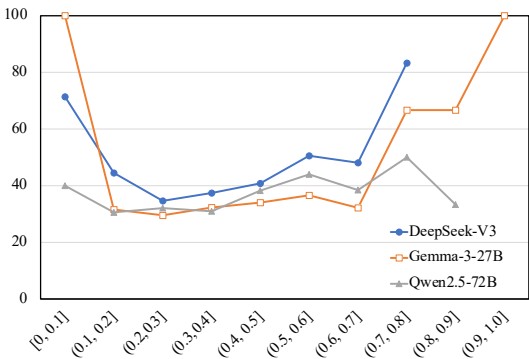

Figure 13: Micro-accuracy(%) of LLMs performance under different compression ration with corresponding tokenizers.

```
(q̄\_/p̄)
 /. .\. .-"""-.
=\_t_/=    /   `\    ___
  )\ ))___,_\    `\  |___)
 nn-nn`   `nn---'
```

What is depicted in the above ASCII art?
A. monkey          B. fish
C. mouse           D. butterfly

(a) The original case from ASCIIEVAL

```
(¯q_¯p)
/ . . . . =¯t/=  -"""- .
(¯
nn-nn)___(¯nn-¯nn)
```

(b) ASCII art repeated by Qwen2.5-VL-72B

Figure 14: Case study for Qwen2.5-VL-72B. The choices in red and in blue refer to the correct answer and the incorrect candidate chosen by the model, respectively.

This reply confirms that the model is not *seeing* a mouse at all. Instead, its OCR abilities lead it to interpret the art as a "mathematical or logical expression," a clear failure of holistic visual reasoning.

Furthermore, when we asked the model to simply list the characters it perceived, it successfully extracted many of them. However, their spatial arrangement in the output was completely disordered. This reinforces the conclusion that while the model excels at identifying characters, it is challenging to process their positional relationships.

## P  CASE STUDIES

We selected seven samples belonging to different classes from ASCIIEVAL and show the cases in Fig. 15 and Fig. 16. The correct answers are marked in red.

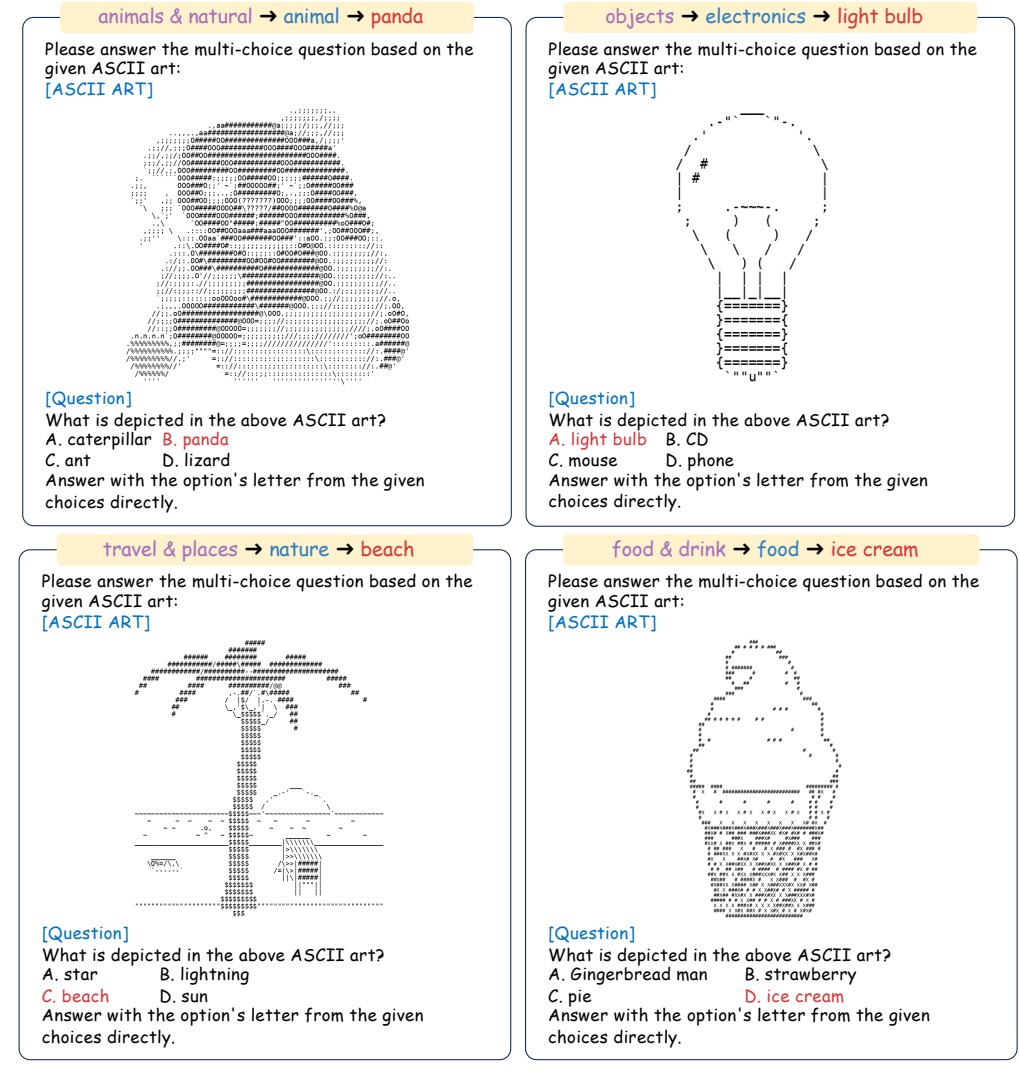

Figure 15: Case studies (Part I).

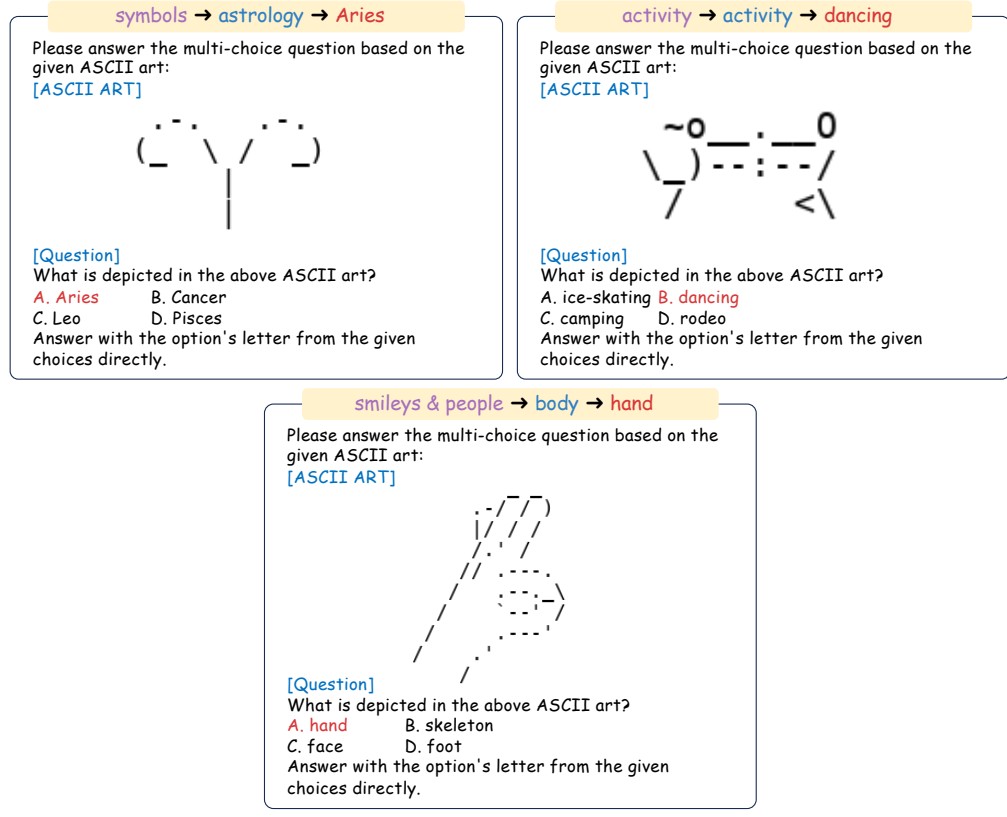

Figure 16: Case studies (Part II).

