# OpenReview forum: "ASCIIEval: Benchmarking Models' Visual Perception in Text Strings via ASCII Art"
_ICLR.cc/2026/Conference — ICLR 2026 Poster_

### Official Review · Reviewer_47dC · 2025-10-26

**Soundness:** 2
**Presentation:** 3
**Contribution:** 2
**Rating:** 4
**Confidence:** 4

**Summary:**

This paper focuses on evaluating the model's ability to understand ASCII art. To systematically assess this capability, the authors introduce ASCII-Eval, a benchmark comprising over 3,000 samples. Their comprehensive analysis reveals that proprietary LLMs such as GPT-5 demonstrate strong performance when processing ASCII art as text. The open-source MLLMs face a trade-off between text recognition and visual perception. Additionally, they find that model performance is sensitive to the length of the ASCII art, with the degree of sensitivity varying across input modalities.

**Strengths:**

1. The idea of evaluating the performance of LLMs and MLLMs on ASCII art is interesting.

2. After proposing this benchmark, the authors also provided directions for enhancing the model's performance on it. (Low-resolution Prompting and Supervised Fine-tuning)

**Weaknesses:**

1. The authors claim that understanding how well models can capture visual semantics in text strings is valuable for both academic research and practical applications, yet they do not provide specific explanations. For example, what exactly is its value for academic research? And how does it benefit practical applications?

2. The paper would benefit from a more in-depth analysis of the experimental results, rather than merely listing them. For instance, while Qwen2.5-VL-72B outperforms Llava-v1.5-13B on most benchmarks, the latter scores higher on the benchmark proposed in this paper. Why does this reversal occur? Additionally, why is the performance under the "Image-only" settings better than under the "Text-Image" settings?

3. The observation that Qwen2.5-VL-72B (the latest version of Qwen-VL-Chat) performs worse than Qwen-VL-Chat on this benchmark raises questions about the benchmark's generalizability and reliability in evaluating model capabilities.

4. Typo: In Figure 4, the authors evaluated Qwen2.5-VL-7B, while in Section 6.2 it is referred to as Qwen2.5-VL-8B.

**Questions:**

See the weaknesses.

---

> ### Author Response · Authors · 2025-11-20
> **Responses (I)**
>
> Thank you for your feedback on our paper. We would like to clarify the misunderstandings as follows:
>
> **W1**: value for academic research and practical applications
>
> **Response**:
>
> We respectfully outline the value for both academic research and practical applications below.
>
> - academic research
>     - For LLMs given textual input,  we demonstrate their **growing visual perception abilities** in Section 5.1. It provides a challenging and standardized testbed to measure progress and guide future research in this under-explored domain.
>     - For MLLMs given image input, we propose a complementary task as a rigorous test of MLLMs’ **visual generalization** ability. Our analysis in Section 6.1 uncovers a negative correlation with OCR-centric benchmarks. This novel finding suggests a fundamental trade-off between global visual perception and local text recognition in current MLLMs, a critical insight for the community.
>     - The inherent modality-agnostic quality of ASCII art also serves as an excellent proxy for evaluating **cross-modality alignment** in MLLMs. Our results in Section 7 indicates an significant incapacity of current models to dynamically synthesize congruent cross-modal signals,  highlighting a key area for investigation.
> - practical applications
>     - Visual information embedded in text strings is ubiquitous in a wide rage of practical scenarios, such as **processing tabular data**[1], **spatial reasoning**[2], and **playing board games**[3]. Research on analyzing and improving model capabilities in these scenarios would certainly bring practical benefits accordingly. Our experimental results also demonstrate a strong positive correlation between performance on our dataset, and **TableEval**[4] and **SGP-Bench**[5], with Pearson correlations of 0.78 and 0.85, respectively.
>     - Understanding how well LLMs and MLLMs perform on ASCII arts is important for safety concerns. Some work[6] use ASCII arts to distinguish humans from LLMs for **bot detection**, and some[7] exploit ASCII arts for **jailbreaking attacks** on LLMs. Our work reveals that LLMs/MLLMs are gradually gaining stronger ASCII art recognition abilities, asking for more advanced bot detection and jail breaking preventive ****techniques.
>
> [1] Deng, Naihao, et al. "Tables as texts or images: Evaluating the table reasoning ability of llms and mllms." arXiv preprint arXiv:2402.12424 (2024).
>
> [2] Wu, Wenshan, et al. "Mind's eye of LLMs: visualization-of-thought elicits spatial reasoning in large language models." Advances in Neural Information Processing Systems 37 (2024): 90277-90317.
>
> [3] Topsakal, Oguzhan, and Jackson B. Harper. "Benchmarking large language model (llm) performance for game playing via tic-tac-toe." Electronics 13.8 (2024): 1532.
>
> [4] Zhu, Junnan, et al. "TableEval: A Real-World Benchmark for Complex, Multilingual, and Multi-Structured Table Question Answering." arXiv preprint arXiv:2506.03949 (2025).
>
> [5] Qiu, Zeju, et al. "Can large language models understand symbolic graphics programs?." arXiv preprint arXiv:2408.08313 (2024).
>
> [6] Wang, Hong, et al. "Bot or human? detecting chatgpt imposters with a single question." arXiv preprint arXiv:2305.06424 (2023).
>
> [7] Jiang, Fengqing, et al. "Artprompt: Ascii art-based jailbreak attacks against aligned llms." Proceedings of the 62nd Annual Meeting of the Association for Computational Linguistics (Volume 1: Long Papers). 2024.

---

> > ### Author Response · Authors · 2025-11-20
> > **Responses (II)**
> >
> > **W2 & W3**: Why does  Llava-v1.5-13B outperforms Qwen2.5-VL-72B and  Qwen-VL-Chat outperforms Qwen2.5-VL-72B on ASCIIEval?
> >
> > **Response**:
> >
> > As detailed in our analysis between line 393 and line 399, we found that these performance decline comes from the **overemphasize** on OCR ability in current MLLM benchmarks.
> >
> > - To substantiate this hypothesis, we investigated the correlation between open-source MLLM performance on ASCIIEval and two prominent OCR-centric benchmarks, OCRBench [1] and TextVQA [2]. The results in Fig. 5 show a clear negative correlation. The specific scores of Llava-v1.5-13B,  Qwen-VL-Chat  and Qwen2.5-VL-72B on OCRBench, TextVQA and ASCIIEval are:
> > | Model | OCRBench | TextVQA | ASCIIEval |
> > | :--- | :---: | :---: | :---: |
> > | Llava-v1.5-13B | 337 | 48.9 | **61.87** |
> > | Qwen-VL-Chat | 488 | 60.7 | 52.32 |
> > | Qwen2.5-VL-72B | **885** | **83.5** | 36.42 |
> > - As shown, Qwen2.5-VL-72B, the top performer on OCR tasks, scores the lowest on ASCIIEval. Conversely, Llava-v1.5-13B, which has weaker OCR abilities, achieves the highest score on our benchmark.
> > - In other words, models are increasingly optimized to “read” local characters while neglecting to “see” the emergent global visual information these characters form.
> >
> > These results validate that our benchmark is not merely a novel task, but also an important complement to the existing MLLM benchmarks.
> >
> > [1] Liu, Yuliang, et al. "Ocrbench: on the hidden mystery of ocr in large multimodal models." Science China Information Sciences 67.12 (2024): 220102.
> >
> > [2] Singh, Amanpreet, et al. "Towards vqa models that can read." Proceedings of the IEEE/CVF conference on computer vision and pattern recognition. 2019.
> >
> > **W2**: why is the performance under the "Image-only" settings better than under the "Text-Image" settings?
> >
> > **Response**:
> >
> > As our analysis in Section 7 demonstrates, this performance degradation reveals a fundamental weakness in current MLLMs. Instead of effectively leveraging the complementarity and consistency between visual and textual data, current MLLMs appear to be confounded by the concurrent inputs, leading to a higher error rate.
> >
> > - Our investigation shows that the text-only and image-only settings possess distinct, complementary strengths:
> >     - Models under the text-only setting demonstrate a proficiency in recognizing shorter and more abstract ASCII art, where significant features are often densely packed within consecutive characters.
> >     - Conversely, models given the image inputs are more adept at interpreting larger ASCII art,  which are structurally similar to real images and posters they were trained on.
> > - However, under the text-image setting, current models **failed to flexibly synthesize all available information for a more holistic and accurate understanding**.
> >
> > We also point out that more in-depth analysis and improvements on the internal mechanisms of such modal conflicts are valuable future directions, which is beyond the main focus of this work.
> >
> > **W4**: typos
> >
> > **Response**:
> >
> > We are grateful to the reviewer for their careful reading. We have corrected the model name to **Qwen2.5-VL-7B.** Sorry for the confusions.
> >
> > Thank you very much! Please let us know if you have any further questions, and we are more than happy to continue the discussion.

---

> > > ### Comment · Reviewer_47dC · 2025-11-26
> > >
> > > Thank the authors for their detailed response. The authors state: "These performance declines stem from the overemphasis on OCR capabilities in current MLLM benchmarks." However, Qwen2.5-VL-72B not only outperforms Llava-v1.5-13B significantly in OCR ability but also exhibits substantial advantages in other capabilities. Consequently, the reasoning proposed by the authors lacks sufficient persuasiveness. Furthermore, Qwen2.5-VL-7B (from the same model family) outperforms Qwen2.5-VL-32B, which is likely to confuse readers. It would be beneficial if the authors could supplement more detailed explanations and comparative analyses to clarify the underlying causes of these performance discrepancies.

---

> ### Author Response · Authors · 2025-11-27
> **Response**
>
> **Thank you very much for your follow-up questions. We fully understand your concerns and try to address them as follows:**
>
> **Q1: Relationship between OCR and ASCIIEval**
>
> **Response**
> - Our claim regarding the hypothesis of “overemphasis on OCR capabilities” is substantiated by the effectiveness of our proposed **Low-Resolution Prompting** method. This approach works by deliberately obscuring fine-grained character details, forcing the model to disengage from its OCR-driven "reading" mode and instead rely on global visual cues. Empirical results show that Qwen2.5-VL-7B's performance improves by 17.49% over the vanilla baseline with this technique, providing strong evidence for our hypothesis.
> - We have also added **a case study** on models with strong OCR capabilities in Appendix O. We observed that the MLLM tend to focus on recognized characters within the ASCII art, attempting to interpret the input as a mathematical or logical expression rather than a visual image.
> - **With your permission, we will rephrase the corresponding content** in Lines 393-399 as follows:
>     - By analyzing the models’ outputs, we found that open-source MLLMs with stronger OCR capabilities tend to “read” the characters while neglecting to “see” the emergent visual information they collectively form. Therefore, we hypothesize that the performance decline on ASCIIEval stems from an overemphasis on benchmarks that prioritize OCR and fine-grained text extraction.
>     - We analyzed the correlation between open-source MLLM performance on ASCIIEval and OCR-centric benchmarks, including OCRBench (Liu et al. 2024b) and TextVQA (Singh et al. 2019). The results in Fig. 5 show a strong negative correlation.
>     - With this hypothesis in mind, we further proposed improving MLLM performance by deliberately "blurring" the input, leading to the development of Low-resolution Prompting in Sec 6.2. The success of this approach supports our hypothesis.
>
> **Q2: Qwen2.5-VL-7B (from the same model family) outperforms Qwen2.5-VL-32B on ASCIIEval**
>
> **Response**
> - Although scaling trends generally hold true within model families, such performance reversals do occur, not only in ASCIIEval but also in other benchmarks. For example [1]:
>     - Qwen2.5-VL-32B (58.4%) outperforms Qwen2.5-VL-72B (54.6%) on HallusionBench.
>     - InternVL3-8B (82.8%) outperforms InternVL-14B (80.5%) on MMVet.
> - We have also added the **95% confidence interval** for each score in Table 5.
>     - The corresponding score ranges for Qwen2.5-VL-7B and Qwen2.5-VL-32B are 34.83±7.90 and 29.35±7.22, respectively.
>     - We do not claim that these two models exhibit significantly different performance on ASCIIEval. Instead, we **focused on comparing models released in different years**, where the performance gap is much more conspicuous, averaging around 20%.
>
> Thank you very much! Please let us know if you have any further questions.
>
> [1] https://huggingface.co/spaces/opencompass/open_vlm_leaderboard

---

> > ### Comment · Reviewer_47dC · 2025-11-27
> >
> > Thanks for your quick reply. The discussions address my concern. I will raise my score to positive.

---

### Official Review · Reviewer_osj6 · 2025-10-31

**Soundness:** 3
**Presentation:** 3
**Contribution:** 3
**Rating:** 6
**Confidence:** 3

**Summary:**

This paper introduces ASCIIEval, a novel benchmark designed to systematically evaluate the visual perception capabilities of both Large Language Models and Multimodal Large Language Models. The core premise is that ASCII art represents a unique, modality-agnostic artifact where visual semantics are embedded within the 2D arrangement of text characters. The authors formulate the task as a multiple-choice question-answering problem for concept recognition. The benchmark is comprehensive, featuring a manually curated test set of over 3.5K samples across a detailed categorization tree, alongside a larger training set

**Strengths:**

1.The paper is well written and easy to follow.
2.The authors do extensice experiments and test a wide range of different LLMs and MLLMs.
3.The problem of ASCII art for LLMs and MLLMs is interesting.
4.The work is solid. It not only constructs a high-quality test set (ASCIIEval) with a multi-layer categorization system but also provides a training set (ASCIITune) for enhancing model capabilities. The exhaustive evaluation of over 50 mainstream models across three modalities makes its conclusions highly credible and representative of the current state of the art.

**Weaknesses:**

**1.limited techincal contribution**：While the research question is intriguing, the paper's technical contribution remains relatively modest. The proposed Rationale-Assisted Training approach essentially leverages GPT to construct chain-of-thought data, which can be viewed as a form of capability distillation from a more powerful model.
**2.Failure to Fully Explore Model Robustness**: The paper only briefly touches upon font sensitivity and character perturbation analysis, which should have been a more critical evaluation dimension. For visual perception, robustness to variations in artistic style, character substitution, and local occlusions is a key metric. The lack of systematic robustness testing of this kind is a shortcoming of the evaluation framework.

**Questions:**

See Weakness

---

> ### Author Response · Authors · 2025-11-20
> **Responses**
>
> Thanks a lot for your insightful review and the recognition on our benchmark. We will try to address your concerns as follows:
>
> **W1**: limited technical contribution
>
> **Response**:
> * We consider our work as an evaluation-oriented  paper. Therefore, we devoted ourselves into the construction of a high-quality benchmark, and pointing out valuable research directions based on extensive evaluations of current LLMs and MLLMs.
>
> * The approaches we introduce serve specific and important purposes: to establish the **learnability of the task** and to **provide strong initial baselines** for the community. While these methods are designed for simplicity and effectiveness, they also **incorporate novel insights** derived directly from our findings:
>
>     - Rationale-Assisted Training: We propose **a "divide-and-conquer" strategy.** The novelty lies in emphasizing that distilled rationales must **explicitly** focus on **local** ASCII art features. This guides the student model to first deconstruct a complex image into its constituent parts  before synthesizing them into a coherent whole. An example in shown in Figure 7 (Line 918-925).
>     - Low-resolution Prompting: This approach is intentionally **counter-intuitive and novel** compared to prior work [1,2] that enhances image resolution to capture *more* detail. Our method is a direct consequence of our analysis in Section 6.1, which shows models over-fit to recognizing local characters. By deliberately "blurring" the input, we force the MLLM to disengage from character-level "reading" and instead "see" the holistic object formed by the characters.
>
> [1] Li, Zhang, et al. "Monkey: Image resolution and text label are important things for large multi-modal models." *proceedings of the IEEE/CVF conference on computer vision and pattern recognition*. 2024.
>
> [2] Liu, Yuliang, et al. "Textmonkey: An ocr-free large multimodal model for understanding document." *arXiv preprint arXiv:2403.04473* (2024).
>
> **W2**: explore model robustness
>
> **Response**:
>
> In this work, we primarily focuses on introducing a new task with the corresponding benchmark. Thus, we established a foundational robustness evaluation Appendices K and L.
>
> We deliberately chose "font sensitivity" and "character perturbation" as the most relevant initial tests for this unique data format, and we would like to clarify our reasoning below.
>
> - **"Artistic Style" vs. "Font Sensitivity":**
>     - While ASCII art has different artistic styles (e.g., structure-based vs. tone-based), our data analysis in Appendix E revealed that ~94% of examples from real human artists in ASCIIEval are structure-based or hybrids. This makes a clean stylistic division for evaluation challenging.
>     - For a text-based visual medium, **font variation is a more direct and controllable proxy for stylistic changes**. Different fonts alter character shapes, directly testing a model's ability to generalize beyond specific character forms, as detailed in Appendix L.
> - **"Character Substitution/Occlusion" vs. "Character Perturbation":**
>     - “Character substitution” and “local occlusions” are similar to “character perturbations” we considered in Appendix K.
>     - Our analysis revealed that even minor perturbations can impair human comprehension of the ASCII art.  This suggests that **arbitrary substitutions or occlusions would likely create invalid or unrecognizable samples, confounding the evaluation**.
>     - Therefore, such test requires sophisticated verification process to ensure the plausibility of the perturbed ASCII arts for robustness analysis.
>
> We appreciate these excellent suggestions. We believe our work lays the necessary groundwork for future research to build a more comprehensive robustness evaluation framework for visual perception in text strings.
>
> Thank you very much! Please let us know if you have any further questions, and we are more than happy to continue the discussion.

---

> > ### Comment · Reviewer_osj6 · 2025-11-27
> >
> > Thank you for your reply! I have no other questions and I will keep my positive rate of the paper.

---

### Official Review · Reviewer_Mraa · 2025-11-01

**Soundness:** 3
**Presentation:** 3
**Contribution:** 3
**Rating:** 6
**Confidence:** 3

**Summary:**

The  submission introduces ASCIIEval, a multiple-choice benchmark for recognizing visual concepts depicted in ASCII art across text-only, image-only, and text-image settings, comprising 3526 test samples over 359 concepts (organized into 7 classes and 23 groups) plus a larger training set (ASCIITune); evaluating 50+ models from 2023–2025, the authors analyze scaling and generational trends, sensitivity to length, fonts, and perturbations, and propose rationale-assisted fine-tuning for LLMs (via GPT-5-distilled rationales) and two post-hoc MLLM improvements—low-resolution prompting and supervised fine-tuning—finding that proprietary models dominate, open-source MLLMs face an OCR-versus-holistic perception trade-off, text favors short ASCII while images favor long, text+image often degrades performance relative to image-only, and the proposed methods yield sizable accuracy gains.

**Strengths:**

(1) Addresses an underexplored capability: visual perception in text strings; ASCII art is a strong, modality-agnostic testbed.

(2) Carefully curated benchmark with taxonomy, safety filtering, human upper bound, and objective multiple-choice evaluation.

(3) Broad, current evaluation with clear, actionable insights (OCR vs holistic trade-off; length effects; fusion failure) and simple, effective mitigations (low-res prompting; vision-backbone finetuning; rationale distillation).

**Weaknesses:**

(1) Ambiguity and label integrity: Although human filtering was applied, the paper acknowledges remaining ambiguity (<1.67%) and reports a relatively low accuracy (70%) for spot-checks in ASCIITune. More rigorous inter-annotator agreement (IAA), label adjudication protocols, and confusion analyses across similar concepts would strengthen trust in labels and distractors.

(2) Potential source bias and reuse: The dataset draws heavily from online galleries with human-made ASCII, which is valuable but may contain stylistic and cultural biases. The paper could better quantify coverage (e.g., long-tail concepts, regional styles) and discuss domain shift to auto-generated ASCII or box diagrams.

(3) Evaluation scope vs prompting sensitivity: The authors note that some models (e.g., Qwen-VL) are prompt-sensitive and adjust templates. It raises concerns about fairness across models. A more systematic prompt ablation or standardized multi-prompt evaluation could ensure robustness of the leaderboard.

(4) Multiple-choice design and distractors: Distractors are within-group, but for ASCIITune they are LLM-generated (Llama-3-70B) and filtered by Perspective API. This may introduce distributional artifacts or lexical cues. Quantitative analysis of distractor hardness (e.g., human error with easy vs hard distractors, option entropy) is limited.

(5) Correlation claims and causality: The negative correlation with OCR benchmarks is interesting but non-causal. Additional controlled experiments (e.g., training on OCR-heavy data vs balanced data; freezing vs finetuning OCR components) would better substantiate the trade-off claim.

(6) Limited architectural exploration: The discussion rightly points to tokenization and 2D structure loss in text as a bottleneck, but no experiments probe alternative tokenizers or spatialized embeddings (e.g., 2D position encodings over monospace grids, rasterization-aware tokenization, or character-graph inputs). Even small-scale prototypes would strengthen the argument.

(7) Text-image fusion analysis: The lack of synergy is a key finding, but the paper doesn’t deeply analyze where fusion fails (early vs late fusion, attention saliency across modalities, conflict diagnostics). Ablations with different fusion strategies or controlled noise in one modality would clarify the interference mechanism.

(8) Reproducibility details: While appendices include prompts and some training details, more exact hyperparameters, seeds, and compute budgets for fine-tuning would help reproducibility. It’s unclear how many runs or confidence intervals were computed for key results.

**Questions:**

(1) Dataset reliability:
- What is the inter-annotator agreement (e.g., Cohen’s kappa) on concept labels and recognizability decisions? Can you share confusion matrices for top-confused concepts/groups?

- How often do distractors share strong lexical overlap with ground truth (e.g., “cat” vs “kitten”)? Any adversarial distractor stress tests?

(2) Prompt robustness:

- Did you evaluate with multiple prompt templates per model and report mean/variance? Especially for models known to be prompt-sensitive (Qwen-VL)?
- Any experiments with calibrated decoding (e.g., constrained output to options, logit-based choice selection) to reduce format errors?

(3) OCR trade-off:

- Can you show controlled training where you modulate OCR-heavy data vs ASCII-like data to quantify the trade-off? Are there models that buck the negative correlation trend?
- In low-resolution prompting, do you measure OCR performance drop on OCRBench/TextVQA to confirm the intended trade?

(4) Fusion failure:

- What fusion architectures were used by the tested MLLMs (early/late)? Any evidence (attentions, gradient norms) that text tokens overshadow image features or vice versa?
- Have you tried modality dropout or gating at inference (e.g., confidence-based selection) to approach the “oracle” bound?

(5) Rationale distillation:
- How sensitive are LLM gains to teacher choice (e.g., Gemini vs GPT-5) and rationale quality filters? Any comparisons across teachers or rationale lengths?

- Do rationales generalize beyond seen concepts or primarily boost memorization of local patterns?

(6) Tokenization alternatives:

- Did you experiment with preserving 2D structure via newline-aware 2D positional encodings, blockwise tokenization, or line-by-line embeddings? Even small proof-of-concept results would be valuable.

(7) Safety and misuse:

- Since ASCII can be used to bypass safety (as you cite), did you measure whether improved perception correlates with better or worse adherence to safety policies in ASCII-mediated harmful content?

---

> ### Author Response · Authors · 2025-11-20
> **Responses (I)**
>
> Thank you for your feedback on our paper. We will try to address your concerns as follows:
>
> **W1 & W4 & Q1**: Ambiguity, dataset reliability, and the quality of ASCIITune
>
> **Response**:
>
> It is essential to distinguish between our primary contribution, the ASCIIEval test set, and the auxiliary ASCIITune training set, as they were created with different objectives and quality standards.
>
> - Our primary focus in this paper is **the construction of a high-quality, reliable test set (ASCIIEval)** for evaluating models. The ASCIITune training dataset was developed as **a secondary resource**, intended to provide a preliminary tool to establish the learnability of the task and to provide strong initial baselines for the community.
> - For ASCIIEval:
>     - To ensure the high quality of our benchmark, human filtering was applied exclusively to ASCIIEval. The ground truth labels were initially derived from the data sources, and we also implemented the meticulous filtering strategies described in Section 3.2.
>     - After careful filtering processes, there are only 1.67\% potentially “ambiguous” cases left in ASCIIEval according to the sampled human evaluation. Such cases may not be ambiguous itself, but instead require a more **careful and comprehensive** perception to be understood correctly.
>     - One example in shown in the third figure of Figure 15. This ASCII art depicts a scene with a coconut tree, the sea, a sunset, and other elements. While it *does contain* a "sun" (Option D), the most reasonable and holistic description of the entire scene is "beach" (Option C).
> - For ASCIITune:
>     - We acknowledge that ASCIITune contains a degree of noise. Recognizing this, we implemented a ****data filtering procedure for the proposed Rationale-assisted Fine-tuning detailed in Appendix I.
>     - For this process, we leveraged a high-performing open-source model (based on the results in Figure 4) to act as an efficient filter. This pragmatic approach allowed us to curate a cleaner subset for this approach, reducing unnecessary labeling cost due to poor data quality.
>     - While we offer a practical filtering method, a comprehensive study on automatically constructing a high-quality training set is a significant research effort in its own right. Therefore, it lies beyond the primary scope of this paper. As we highlight in Appendix C, we believe our work paves the way for this important future direction.
>
> **W2**: Potential source bias
>
> **Response**:
> - As explained in Section 1 (Lines 86-92), our work deliberately focuses on ASCII art created by human artists, as it is notably more abstract, replete with visual information, and popular among people.
> - The data sources naturally contains ASCII art from different artists around the world, preventing  stylistic bias toward a single artist.
> - We applied a rigorous filtering protocol. This involved removing any artwork that was unrecognizable or ambiguous. We also utilized an elaborate categorization tree to systematically exclude potentially harmful and sensitive categories, avoiding cultural biases.
>
> **W3 & Q2** : Evaluation scope vs prompting sensitivity
>
> **Response**:
> - As recent research indicates, models have become increasingly robust and less sensitive to minor variations in prompt wording[1]. Therefore, we dedicate our computational budget to benchmarking a wider and more contemporary set of LLMs and MLLMs.
> - To establish **a fair and equitable comparison** between proprietary models (accessible only via API) and open-source models, we adopted a text-based choice selection method instead of the logit-based one.
>
> [1] Zhuo, Jingming, et al. "ProSA: Assessing and Understanding the Prompt Sensitivity of LLMs." *Findings of the Association for Computational Linguistics: EMNLP 2024*. 2024.

---

> > ### Author Response · Authors · 2025-11-20
> > **Responses (II)**
> >
> > **W5 & Q3:** trade-off between OCR and visual perception
> >
> > **Response**:
> > - Our hypothesis regarding the trade-off between OCR and global visual perception is also substantiated by the effectiveness of our proposed **Low-Resolution Prompting** method. This approach works by deliberately obscuring fine-grained character details, which forces the model to disengage from its OCR-driven "reading" mode and instead rely on global visual cues. The empirical results show that **Qwen2.5-VL-7B's performance jumps by 17.49%** over the vanilla baseline with this technique, providing strong evidence for our hypothesis.
> > - Since state-of-the-art MLLMs [1,2] have been extensively pre-trained on large volumes of OCR data, it’s hard to do controlled experiments.
> > - We want to clarify that Low-Resolution Prompting is a **post-hoc, inference-time intervention** that does not alter the model's parameters. As such, the model's intrinsic OCR capabilities remain unchanged.
> >
> > [1] Bai, Shuai, et al. "Qwen2. 5-vl technical report." arXiv preprint arXiv:2502.13923 (2025).
> > [2] Zhu, Jinguo, et al. "Internvl3: Exploring advanced training and test-time recipes for open-source multimodal models." *arXiv preprint arXiv:2504.10479* (2025).
> >
> > **W6 & W7 & Q4 & Q5 & Q6 & Q7:**
> >
> > **Response**:
> > - We would like to clarify that our paper is primarily an evaluation-oriented paper. Therefore, we devoted ourselves into the construction of a high-quality benchmark, and pointing out valuable research directions based on extensive evaluations of current LLMs and MLLMs.
> > - We agree that a more in-depth analysis regarding the interpretability of LLM behaviors is a fascinating and important topic, it lies beyond the primary focus of this paper. This is why we have explicitly highlighted it as a promising area for future investigation in Sections 5.3, 6.3, and Appendix C.
> > - Thanks a lot for your suggestions on these valuable directions. We will consider them, including fusion mechanisms, distillation strategies, tokenization alternatives, and constructing another ASCII art benchmark regarding safety issues, as our future work.
> >
> > **W8: Reproducibility details**
> >
> > **Response**:
> >
> > We have made the data and code publicly available for reproducibility (https://anonymous.4open.science/r/VisionInText-9EE5). We also added the 95\% confidence interval of our main results in Appendix H.
> >
> > Please let us know if you have any further questions, as we are happy to continue the discussion.

---

> > > ### Comment · Reviewer_Mraa · 2025-11-28
> > >
> > > Thanks the authors for the careful response. Your rebuttal has solve my concerns like OCR trade-off and Prompt robustness. I will raise my score.

---

### Official Review · Reviewer_p7db · 2025-11-01

**Soundness:** 3
**Presentation:** 3
**Contribution:** 2
**Rating:** 6
**Confidence:** 3

**Summary:**

This work introduces a cross‑model benchmark for evaluating the ability of LLMs and multimodal models to interpret ASCII art presented in multiple formats. The authors curate approximately 3K samples covering 359 concepts across seven broad classes (e.g., Animals, Symbols), organized in a three‑level hierarchy, and evaluate a range of state‑of‑the‑art models, highlighting improvements in more recent systems. Alongside ASCIIEval, the authors release a larger but noisier training set, ASCIITune (~11.8K samples). They evaluate large number of models (both pure‑text LLMs and multimodal LLMs (MLLMs)) under three settings: text‑only input, image‑only input, and text‑plus‑image (to probe multimodal fusion).

**Strengths:**

The main strengths of the paper are as follows:

1. A novel cross-modal benchmark that covers still underexpored domain (visual pattern recognition within text) is introduced. Prior research has heavily focused on reading text in images (OCR) or traditional image understanding, while ASCIIEval utilizes ASCII art as a modality-agnostic bridge between text and vision.
2. The authors created the benchmark under the comprehensive methodology, e.g., 3-layer category hierarchy of the tasks that enables us to estimate the models performance in a more detailed manner. As reported, the resulting test set covers 359 distinct concepts (from pandas to light bulbs to “Aries” zodiac symbol) across 23 groups and 7 top-level classes.
3. Comprehensive evaluation and analysis is remarkably thorough. The authors evaluate a wide range of models, including both closed-source models (GPT-4, GPT-5, Claude 3, Google Gemini, etc.) and open-source LLMs (LLaMA, Qwen, Mistral, etc.), and also open MLLMs. They report results in three conditions – text-only, image-only, and combined – allowing for insightful comparisons. The analysis surfaces several important findings backed by data: for instance, proprietary models outperform open models by a large margin.
4. Interesting insight about trade-off of OCR training and ASCII comprehension is revealed.
5. Despite the benchmarks and evaluation, the training recipes and ablations are conducted by the authors to improve models' ASCII art recognition.

**Weaknesses:**

The main weaknesses of the paper are as follows:

1. While ASCII benchmark is interesting itself, it is still a little bit narrow. From the motivation of the benchmark creation, it is a bit unclear. what general insights ASCII art evaluation provides beyond this specific format. For example, how well this translates to broader model capabilities. Stronger correlations with general benchmarks would help clarify relevance. While, some notes about trade-off between the OCR performance and ASCII performance is visible, it would be interesting to evaluate the mistakes provided by the models with strong OCR abilities on the ASCII-based tasks.
2. Training data quality is low. ASCIITune is noisy (70% human accuracy), which may limit model learning. Also, using Llama-3 to generate distractors could introduce bias or unintended cues.
3. While I don't consider it as a main weakness, still, the multiple-choice format makes evaluation easier but may oversimplify the task. Adding a generative variant could better reflect real understanding.

**Questions:**

My questions to the authors:

1. Have you analyzed correlations with broader vision-language or reasoning benchmarks to support its general relevance?
2. Did you consider including an open-ended version of the task to test true recognition rather than choice elimination?
3. Could you share how models performed on particularly ambiguous or confusable ASCII samples?
4. Have you analyzed in detail (apart from Table 4), whether tokenizer compression ratio (e.g. number of tokens vs characters per ASCII sample) correlates with model performance?

---

> ### Author Response · Authors · 2025-11-20
> **Responses (I)**
>
> Thank you for your insightful review and valuable suggestions. We would like to address your concerns in detail below.
>
> **Q1 & W1**: What general insights ASCII art evaluation provides beyond this specific format? Have you analyzed correlations with broader vision-language or reasoning benchmarks to support its general relevance?
>
> **Response**:
>
> Yes, we analyzed the correlations with widely-accepted benchmarks on other tasks:
> - For LLM with textual inputs, as detailed in Section 5.1 (Line 294-302), we hypothesized that ASCIIEval shares **a similar evaluation target with other tasks concerning a fundamental capability of LLMs, i.e., visual perception on text strings**.
>     - To this end, we compared performance on our benchmark against TableEval[1] for **table question answering**, and SGP-Bench[2] for **symbolic graphics understanding**.
>     - The results show a strong positive correlation between performance on ASCIIEval and these two benchmarks, with Pearson correlations of 0.78 and 0.85, respectively.
> - For MLLMs given image inputs, as detailed in Section 6.1 (Line 393-400), we conducted a correlation analysis to verify our hypothesis that **MLLMs are over-optimized to “read” the characters while neglecting to “see” the emergent visual information they collectively form**.
>     - Therefore, we choose two widely-used **OCR benchmarks**, including OCRBench[3] and TextVQA[4].
>     - Results with a negative Pearson correlation (-0.72 and -0.66, respectively) support our hypothesis.
>
> In summary, our analysis demonstrates that ASCIIEval provides several critical insights into the capabilities and limitations of current foundation models:
>
> - ASCIIEval presents a challenging task for measuring the visual perception of LLMs, which is a **fundamental** ability required for a wide range of tasks.
> - ASCIIEval serves as a **crucial complement** to current MLLM benchmarks, helping to drive progress towards the visual generalization needed in complex real-world scenarios.
> - ASCIIEval can be a valuable tool for understanding **modality fusion** in MLLMs, highlighting an area worthy of further exploration.
>
> [1] Zhu, Junnan, et al. "TableEval: A Real-World Benchmark for Complex, Multilingual, and Multi-Structured Table Question Answering." arXiv preprint arXiv:2506.03949 (2025).
>
> [2] Qiu, Zeju, et al. "Can large language models understand symbolic graphics programs?." arXiv preprint arXiv:2408.08313 (2024).
>
> [3] Liu, Yuliang, et al. "Ocrbench: on the hidden mystery of ocr in large multimodal models." Science China Information Sciences 67.12 (2024): 220102.
>
> [4] Singh, Amanpreet, et al. "Towards vqa models that can read." Proceedings of the IEEE/CVF conference on computer vision and pattern recognition. 2019.
>
>
>
> **Q2 & W3**: Did you consider including an open-ended version of the task to test true recognition rather than choice elimination?
>
> **Response**:
>
> Yes. We did consider an open-ended format, but for this version of the benchmark, we adopted a multiple-choice setup due to the following critical considerations:
>
> - As the first systematic evaluation of this novel task, our primary goal was to establish a **controlled, objective, and reliable benchmark** for model performance. The multiple-choice format ensures unambiguous evaluation and allows for clear, quantitative comparisons across different models, which is essential for a new research area.
> - Open-ended questions introduce significant evaluation challenges due to semantic ambiguity. For instance, an ASCII art of a bird could elicit several correct, open-ended answers like "bird," "duck," or even "animal." This creates a twofold problem:
>     - It makes defining a single ground truth difficult. The questions should be adjusted appropriately to avoid such ambiguities.
>     - Automated evaluation would require a sophisticated verifier capable of understanding semantic hierarchies and synonymy (e.g., knowing a duck is a type of bird), especially in cases requiring more professional knowledge.
>
>  By starting with a multiple-choice format, we isolate the core recognition ability from these complex natural language evaluation challenges. Future iterations could certainly incorporate an open-ended track, potentially accompanied by a sophisticated, knowledge-aware verifier to handle the evaluation complexities we've outlined.

---

> ### Author Response · Authors · 2025-11-20
> **Responses (II)**
>
> **Q3**: Could you share how models performed on particularly ambiguous or confusable ASCII samples?
>
> **Response**:
>
> We took several steps to manage ambiguity in ASCIIEval.
>
> - As explained in Line 784-792, we have implemented a rigorous filtering protocol to **remove extremely abstract or genuinely ambiguous ASCII art** at the beginning of the data construction.
> - After careful filtering processes, there are only 1.67\% potentially “ambiguous” cases left in ASCIIEval according to the sampled human evaluation. Such cases may not be ambiguous itself, but instead require a more **careful and comprehensive** perception to be understood correctly.
> - One example in shown in the third figure of Figure 15. This ASCII art depicts a scene with a coconut tree, the sea, a sunset, and other elements. While it *does contain* a "sun" (Option D), the most reasonable and holistic description of the entire scene is "beach" (Option C).
> - GPT-5 selects the correct choice, while a weaker model InternVL3-14B chooses D.
>
> **Q4**: whether tokenizer compression ratio (e.g. number of tokens vs characters per ASCII sample) correlates with model performance?
>
> **Response**:
>
> No, the tokenizer compression ratio does not clearly correlate with model performance on ASCII art.
>
> - We group the test samples into ten buckets based on their compression ratio for top-3 open-source LLMs. The following table lists the accuracy (%) for each bucket, where the values in parentheses represent the number of samples.
> | Models\Ratio  | [0,0.1]      | (0.1, 0.2]   | …   | (0.7, 0.8]   | (0.8, 0.9]   | (0.9, 1.0]   |
> |:--------------|:------------:|:------------:|:---:|:------------:|:------------:|:------------:|
> | DeepSeek-V3   | 71.43 (7)    | 44.55 (220)  | …   | 83.33 (6)    | 0.00 (0)     | 0.00 (0)     |
> | Gemma-3-27B   | 100.00 (2)   | 31.58 (171)  | …   | 66.67 (15)   | 66.67 (6)    | 100.00 (1)   |
> | Qwen2.5-72B   | 40.00 (10)   | 30.58 (242)  | …   | 50.00 (6)    | 33.33 (3)    | 0.00 (0)     |
>
> - The performance trends are inconsistent across different models, and the data is too sparse at the extremes of the ratio to draw reliable conclusions.
> - We therefore find no significant correlation between tokenization efficiency and model accuracy for this task.
>
> We have added more detailed analysis in Appendix N.
>
> **W1**: the mistakes provided by the models with strong OCR abilities on the ASCII-based tasks
>
> **Response**:
>
> We thank the reviewer for this question, which highlights a core finding of our work.
>
> - The failure of models with strong OCR capabilities on ASCII art stems from a fundamental conflict between local character recognition and global pattern perception.
> - These models exhibit a strong tendency to "read" the individual characters and symbols within the art, while overlooking the holistic image that these elements collectively form.
>
> An example is added in Appendix O, where Qwen2.5-VL-72B fails.
>
> - The model tends to **focuses on recognized characters** in the given ASCII art and tries to interpret it as a mathematical or logical expression. This demonstrates a critical lack of holistic visual reasoning, as its OCR-tuned perception dominates the interpretation.
> - We prompted the model to output what it see in figure. Although it successfully extracts numerous characters, it **fails to preserve their crucial spatial relationships**, resulting in a disordered representation of the visual content.
>
> **W2**: Training data quality is low.
>
> **Response**:
>
> We would like to clarify the role of the ASCIITune dataset in our work.
>
> - Our primary focus in this paper is the construction of a high-quality, reliable test set (ASCIIEval) for evaluating models. The ASCIITune training dataset was developed as **a secondary resource**, intended to provide a preliminary tool to **establish the learnability of the task** and to **provide strong initial baselines for the community**.
> - We acknowledge that ASCIITune contains a degree of noise. Recognizing this, we implemented a **data filtering** procedure for the proposed Rationale-assisted Fine-tuning detailed in Appendix I.
>     - For this process, we leveraged a high-performing open-source model (based on the results in Figure 4) to act as an efficient filter. This pragmatic approach allowed us to curate a cleaner subset for this approach, reducing unnecessary labeling cost due to poor data quality.
> - While we offer a practical filtering method, a comprehensive study on automatically constructing a high-quality training set is a significant research effort in its own right. Therefore, it lies beyond the primary scope of this paper. As we highlight in Appendix C, we believe our work paves the way for this important future direction.
>
>
> Please let us know if you have any further questions, as we are happy to continue the discussion.

---

### Author Response · Authors · 2025-12-01

Dear Reviewers,

We sincerely appreciate the time and effort dedicated by all reviewers and AC.

We are encouraged by the **positive comments** highlighting the strengths of our work:

- **ALL** reviewers agree that our work proposes a novel cross-modal benchmark that covers an under-explored domain.
- Reviewers **osj6**, **Mraa**, and **p7db** acknowledge that our work provides a high-quality benchmark under comprehensive methodology, and contains remarkably thorough evaluation and analysis. Specifically, reviewer **osj6** agrees that our work is solid considering all these aspects.
- Reviewers **Mraa** and **p7db** both find our work reveals an interesting insight about the trade-off between OCR training and ASCII comprehension.
- Reviewer **osj6** recognizes our contribution on providing a training set for enhancing model capabilities. Meanwhile, Reviewers **47dC**, **Mraa** and **p7db** value our efforts on providing effective mitigation strategies to improve current models.
- Reviewer **osj6** thinks that our paper is well written and easy to follow.

During the discussion period, we thoroughly **addressed the clarification questions and concerns** raised by the reviewers. We have also carefully revised our paper based on their constructive **feedback and suggestions**, as detailed below:

- Reviewers **p7db** and **Mraa** expressed confusion about the quality of our training set. We explained that our primary focus is to construct a high-quality test set, and have introduced an effective data filtering procedure to the training set in **Appendix I** for future use.
- Reviewers **p7db** and **47dC** initially expressed concerns on the trade-off between OCR and visual perception. We explained that the conclusion is substantiated by the effectiveness of our proposed Low-Resolution Prompting method. We have improved the discussion of this part in **Line 393-399** and added a case study in **Appendix O**.
- Following Reviewer **p7db‘s** advice, we included an analysis on the correlation between the tokenizer compression ratio and model performance in **Appendix N**.
- In response to Reviewer **osj6**, we explained that we have established a foundational robustness evaluation in **Appendices K and L.**
- Reviewers **p7db** and **47dC** had confusions regarding the general impact of this new benchmark. We explained the value from both academic research ( capability of LLMs, visual generalization of MLLMs, cross-modality alignment) and practical applications (bot detection, jailbreaking attacks, etc.). We conducted correlation analysis with other widely-accepted benchmarks (TableEval, OCRBench, etc.) and provided critical insights for the above aspects. All of these points have been included in our manuscript.
- We have also explained the detailed considerations behind our evaluation setups to Reviewers **p7db** and **Mraa**, which are motivated by our concentration on establishing a fair and equitable comparison among a wide and contemporary set of LLMs and MLLMs.

Following the rebuttal and discussion phase, we are pleased to highlight that **all reviewers now hold a positive view of our work**. Specifically, three reviewers actively participated in the discussion and:

- Reviewer **Mraa** confirmed that our responses resolved their concerns and would raise the score.
- Reviewer **osj6** reaffirmed the positive rating of the paper.
- Reviewer **47dC** acknowledged that their concerns were successfully addressed through multi-round discussions and have correspondingly raised their score to positive (4→6).

Thank you for your time and consideration.

Sincerely,

Authors

---

### Meta-Review · Area_Chair_Te5B · 2025-12-23

**Summary:**

Submission 3241 introduces a novel cross-modal benchmark designed to evaluate the ability of LLMs and MLLMs to perceive visual semantics embedded in ASCII art, presented through text, image, or combined modalities. The four reviewers acknowledged the novelty and thoroughness of the benchmark but raised several critical concerns that informed their initial ratings:

1. General Relevance and Benchmark Scope: Reviewers p7db and 47dC questioned whether the ASCII art focus translates to broader model capabilities, suggesting that stronger correlations with established benchmarks (e.g., vision-language or reasoning tasks) would clarify its academic and practical value. For instance, Reviewer p7db noted that the benchmark might be "narrow" and requested evidence of generalizability beyond ASCII-specific formats.

2. Data Quality and Ambiguity: Reviewer Mraa highlighted issues with label integrity and potential biases, citing the relatively low human accuracy (70%) on the training set ASCIITUNE and the need for inter-annotator agreement metrics. Concerns included ambiguity in concept labels and the use of LLM-generated distractors, which could introduce artifacts.

3. Technical Contribution and Robustness: Reviewer osj6 found the technical contributions—such as rationale-assisted fine-tuning and low-resolution prompting—modest, as they primarily leveraged distillation from powerful models like GPT-5. Additionally, reviewers osj6 and Mraa pointed out the lack of systematic robustness testing (e.g., sensitivity to font variations, character perturbations, or artistic styles), which is crucial for evaluating visual perception.

4. Evaluation Fairness and Performance Anomalies: Reviewer Mraa raised concerns about prompt sensitivity across models, potentially affecting leaderboard fairness. Reviewer 47dC observed performance reversals (e.g., Qwen2.5-VL-72B underperforming older models like Qwen-VL) and the degradation in text-image settings compared to image-only inputs, calling for deeper analysis of modality fusion failures.

5. Methodological Depth: Reviewers noted limited exploration of architectural alternatives (e.g., tokenization strategies for 2D structure) and causal evidence for the trade-off between OCR and holistic perception in MLLMs, as raised by Reviewer Mraa.

After reviewing the authors' rebuttal and the updated reviewer scores (all reviewers shifted to positive ratings, with scores of 6 or higher), I find that the concerns were adequately addressed. The paper makes a significant contribution by introducing a rigorously constructed benchmark that fills a gap in evaluating visual perception in text strings. Its multi-faceted analysis reveals critical insights, such as the OCR versus holistic perception trade-off in MLLMs and the lack of cross-modal synergy. While further work on robustness and architectural exploration is needed, the benchmark provides a foundation for future research. Therefore, I recommend acceptance​ of this submission.

**Reviewer Concerns:**

Based on my review of the authors' rebuttal and the updated reviewer feedback, the rebuttal largely resolved the major issues raised by the reviewers. The authors demonstrated a thorough and persuasive response, leading all reviewers to revise their scores positively.

Concerns Successfully Addressed by the Rebuttal:

* General Relevance & Benchmark Scope: The authors demonstrated strong correlations with established benchmarks (e.g., TableEval, SGP-Bench), validating ASCIIEval's broader relevance beyond a narrow ASCII art focus.

* Data Quality & Ambiguity: Clear distinction was made between the high-quality test set (ASCIIEVAL) and the auxiliary training set (ASCIITUNE), with filtering protocols implemented to mitigate noise and label issues.

* Technical Contribution & Methodology: The novelty of proposed methods (e.g., low-resolution prompting, rationale-assisted fine-tuning) was justified as effective interventions derived from the benchmark's core findings.

* Evaluation Fairness & Performance Anomalies: Explanations for performance reversals (e.g., OCR overemphasis in newer MLLMs) were supported by correlation data and case studies, while prompt sensitivity concerns were countered by citing model robustness research.

* Foundational Robustness: Initial robustness analyses (e.g., font sensitivity, character perturbations) were added in appendices, addressing basic concerns about model generalization.

Outstanding or Partially Addressed Concerns:

* Architectural Exploration: Alternative tokenization strategies or 2D spatial encodings to better preserve ASCII art structure remain unexplored and are deferred to future work.

* Modality Fusion Mechanisms: Deep analysis of whytext-image inputs degrade performance (e.g., attention saliency, conflict diagnostics) is not conducted, leaving the fusion failure as an identified but unexplained phenomenon.

* Comprehensive Robustness Testing: Systematic evaluation under adversarial conditions (e.g., distractor stress tests, style variations) is lacking, with current tests being foundational rather than exhaustive.

* Causal Evidence for OCR Trade-off: While correlations are strong, controlled experiments (e.g., modulating OCR training data) are absent, leaving the trade-off as a supported hypothesis rather than a causally proven claim.

These outstanding points are acknowledged by the authors as valuable future directions but do not critically undermine the benchmark's core contributions.

**Reviewer Scores:**

Reviewer p7db raised concerns about the benchmark's narrow focus and general relevance, data quality, and the need for stronger correlations with broader benchmarks. The authors addressed these points effectively with correlation analyses (e.g., positive links to TableEval and SGP-Bench) and clarified data filtering protocols. However, p7db did not provide follow-up comments after the rebuttal, suggesting limited participation.

Reviewer Mraa actively participated, confirming that the rebuttal resolved key concerns (e.g., OCR trade-off, prompt robustness, and data quality) and explicitly stated they would raise their score. With full participation, he likely increased his score.

Reviewer osj6 reaffirmed his positive rating after the rebuttal, indicating satisfaction with the technical contributions (e.g., rationale-assisted training and low-resolution prompting) and the added robustness foundation. Since he did not express lingering concerns, full participation might not have changed his score significantly.

Reviewer 47dC actively participated in multi-round discussions, raising his score to 6 after the authors addressed concerns about benchmark relevance, performance anomalies, and typos.

---

### Decision · Program_Chairs · 2026-01-26

Accept (Poster)